# Towards Better Understanding of Adaptive Gradient Algorithms in Generative Adversarial Nets

**Mingrui Liu**[1][*]**, Youssef Mroueh**[2]**, Jerret Ross**[2]**, Wei Zhang**[2]**, Xiaodong Cui**[2]**, Payel Das**[2]**, Tianbao Yang**[1]
[1] Department of Computer Science, The University of Iowa, Iowa City, IA, 52242, USA
[2] IBM T. J. Watson Research Center, Yorktown Heights, NY, 10598, USA

## Abstract

Adaptive gradient algorithms perform gradient-based updates using the history of gradients and are ubiquitous in training deep neural networks. While adaptive gradient methods theory is well understood for minimization problems, the underlying factors driving their empirical success in min-max problems such as GANs remain unclear. In this paper, we aim at bridging this gap from both theoretical and empirical perspectives. First, we analyze a variant of Optimistic Stochastic Gradient (OSG) proposed in (Daskalakis et al., 2017) for solving a class of non-convex non-concave min-max problem and establish $O(\epsilon^{-4})$ complexity for finding $\epsilon$-first-order stationary point, in which the algorithm only requires invoking one stochastic first-order oracle while enjoying state-of-the-art iteration complexity achieved by stochastic extragradient method by (Iusem et al., 2017). Then we propose an adaptive variant of OSG named Optimistic Adagrad (OAdagrad) and reveal an *improved* adaptive complexity $O\left(\epsilon^{-\frac{2}{1-\alpha}}\right)$, where $\alpha$ characterizes the growth rate of the cumulative stochastic gradient and $0 \leq \alpha \leq 1/2$. To the best of our knowledge, this is the first work for establishing adaptive complexity in non-convex non-concave min-max optimization. Empirically, our experiments show that indeed adaptive gradient algorithms outperform their non-adaptive counterparts in GAN training. Moreover, this observation can be explained by the slow growth rate of the cumulative stochastic gradient, as observed empirically.

## 1 Introduction

Adaptive gradient algorithms (Duchi et al., 2011; Tieleman & Hinton, 2012; Kingma & Ba, 2014; Reddi et al., 2019) are very popular in training deep neural networks due to their computational efficiency and minimal need for hyper-parameter tuning (Kingma & Ba, 2014). For example, Adagrad (Duchi et al., 2011) automatically adjusts the learning rate for each dimension of the model parameter according to the information of history gradients, while its computational cost is almost the same as Stochastic Gradient Descent (SGD). However, in supervised deep learning (for example, image classification tasks using a deep convolutional neural network), there is not enough evidence showing that adaptive gradient methods converge faster than its non-adaptive counterpart (i.e., SGD) on benchmark datasets. For example, it is argued in (Wilson et al., 2017) that adaptive gradient methods often find a solution with worse performance than SGD. Specifically, Wilson et al. (2017) observed that Adagrad has slower convergence than SGD in terms of both training and testing error, while using VGG (Simonyan & Zisserman, 2014) on CIFAR10 data.

GANs (Goodfellow et al., 2014) are a popular class of generative models. In a nutshell, they consist of a generator and a discriminator, both of which are defined by deep neural networks. The generator and the discriminator are trained under an adversarial cost, corresponding to a non-convex non-concave min-max problem. GANs are known to be notoriously difficult to train. In practice, Adam (Kingma & Ba, 2014) is the defacto optimizer used for GAN training. The common optimization strategy is to alternatively update the discriminator and the generator (Arjovsky et al.,

---

[*]Correspondence to `mingrui-liu@uiowa.edu`

2017; Gulrajani et al., 2017). Using Adam is important in GAN training, since replacing it with non-adaptive methods (e.g. SGD) would significantly deteriorate the performance. This paper studies and attempts to answer the following question:

**Why do adaptive gradient methods outperform their non-adaptive counterparts in GAN training?**

We analyze a variant of Optimistic Stochastic Gradient (OSG) in (Daskalakis & Panageas, 2018) and propose an adaptive variant named Optimistic Adagrad (OAdagrad) for solving a class of non-convex non-concave min-max problems. Both of them are shown to enjoy state-of-the-art complexities. We further prove that the convergence rate of OAdagrad to an $\epsilon$-first-order stationary point depends on the growth rate of the cumulative stochastic gradient. In our experiments, we observed an interesting phenomenon while using adaptive gradient methods for training GANs: the cumulative stochastic gradient grows at a slow rate. This observation is in line with the prediction of our theory suggesting improved convergence rate for OAdagrad in GAN training, when the growth rate of the cumulative stochastic gradient is slow.

Since GAN is a min-max optimization problem in nature, our problem of interest is to solve the following stochastic optimization problem:

$$\min_{\mathbf{u}\in\mathcal{U}} \max_{\mathbf{v}\in\mathcal{V}} F(\mathbf{u}, \mathbf{v}) := \mathbb{E}_{\xi\sim\mathcal{D}}\left[f(\mathbf{u}, \mathbf{v}; \xi)\right], \tag{1}$$

where $\mathcal{U}$, $\mathcal{V}$ are closed and convex sets, $F(\mathbf{u}, \mathbf{v})$ is possibly non-convex in $\mathbf{u}$ and non-concave in $\mathbf{v}$. $\xi$ is a random variable following an unknown distribution $\mathcal{D}$. In GAN training, $\mathbf{u}$, $\mathbf{v}$ represent the parameters of generator and discriminator respectively.

The ideal goal for solving (1) is to find a saddle point $(\mathbf{u}_*, \mathbf{v}_*) \in \mathcal{U} \times \mathcal{V}$ such that $F(\mathbf{u}_*, \mathbf{v}) \le F(\mathbf{u}_*, \mathbf{v}_*) \le F(\mathbf{u}, \mathbf{v}_*)$ for $\forall \mathbf{u} \in \mathcal{U}, \forall \mathbf{v} \in \mathcal{V}$.

To achieve this goal, the typical assumption usually made is that the objective function is convex-concave. When $F(\mathbf{u}, \mathbf{v})$ is convex in $\mathbf{u}$ and concave in $\mathbf{v}$, non-asymptotic guarantee in terms of the duality gap is well established by a series of work (Nemirovski & Yudin, 1978; Nemirovski, 2004; Nesterov, 2007; Nemirovski et al., 2009; Juditsky et al., 2011). However, when $F(\mathbf{u}, \mathbf{v})$ is non-convex in $\mathbf{u}$ and non-concave in $\mathbf{v}$, finding the saddle point is NP-hard in general. Instead, we focus on finding the first-order stationary point provided that the objective function is smooth. I.e. we aim to find $(\mathbf{u}, \mathbf{v}) \in \mathcal{U} \times \mathcal{V}$ such that $\nabla_{\mathbf{u}} F(\mathbf{u}, \mathbf{v}) = 0$, $\nabla_{\mathbf{v}} F(\mathbf{u}, \mathbf{v}) = 0$. Note that this is a necessary condition for finding the (local) saddle point.

**Related Work.** Several works designed iterative first-order deterministic (Dang & Lan, 2015) and stochastic (Iusem et al., 2017; Lin et al., 2018) algorithms for achieving the $\epsilon$-first-order stationary point with non-asymptotic guarantee. The goal is to find $\mathbf{x}$ such that $\|T(\mathbf{x})\| \le \epsilon$ or $\mathbb{E}\left[\|T(\mathbf{x})\|\right] \le \epsilon$, where the first-order oracle is defined as $T(\mathbf{x}) = \left[\nabla_{\mathbf{u}} F(\mathbf{u}, \mathbf{v}), -\nabla_{\mathbf{v}} F(\mathbf{u}, \mathbf{v})\right]^\top$ with $\mathbf{x} = (\mathbf{u}, \mathbf{v})$ and the first-order stochastic oracle is the noisy observation of $T$, i.e. $T(\mathbf{x}; \xi) = \left[\nabla_{\mathbf{u}} F(\mathbf{u}, \mathbf{v}; \xi), -\nabla_{\mathbf{v}} F(\mathbf{u}, \mathbf{v}; \xi)\right]^\top$. For instance, Dang & Lan (2015) focuses on the deterministic setting. On the other hand, (Iusem et al., 2017) develops a stochastic extra-gradient algorithm that enjoys $O(\epsilon^{-4})$ iteration complexity. The extra-gradient method requires two stochastic first-order oracles in one iteration, which can be computationally expensive in deep learning applications such as GANs. The inexact proximal point method developed in (Lin et al., 2018) has iteration complexity $O(\epsilon^{-6})$ for finding an $\epsilon$-first-order stationary point [1].

To avoid the cost of an additional oracle call in extragradient step, several studies (Chiang et al., 2012; Rakhlin & Sridharan, 2013; Daskalakis et al., 2017; Gidel et al., 2018; Xu et al., 2019) proposed single-call variants of the extragradient algorithm. Some of them focus on the convex setting (e.g. (Chiang et al., 2012; Rakhlin & Sridharan, 2013)), while others focus on the non-convex setting (Xu et al., 2019). The closest to our work is the work by (Daskalakis et al., 2017; Gidel et al., 2018), where the min-max setting and GAN training are considered. However, the convergence of those algorithms is only shown for a class of bilinear problems in (Daskalakis et al., 2017) and for monotone variational inequalities in (Gidel et al., 2018). Hence a big gap remains between the specific settings studied in (Daskalakis et al., 2017; Gidel et al., 2018) and more general non-convex

---

[1]The result in (Lin et al., 2018) assumes the first-order oracle $T$ is a weakly-monotone operator, which is milder than the Lipschitz-continuity assumption as assumed in Iusem et al. (2017). However, simply applying the Lipschitz-continuity condition in their proof does not change their iteration complexity.

| | Assumption | Setting | IC | PC | Guarantee |
|---|---|---|---|---|---|
| Extragradient (Iusem et al., 2017) | pseudo-monotonicity [3] | stochastic | $O(\epsilon^{-4})$ | $2T_g$ | $\epsilon$-SP |
| OMD (Daskalakis et al., 2017) | bilinear | deterministic | N/A | $T_g$ | asymptotic |
| AvgPastExtraSGD (Gidel et al., 2018) | monotonicity | stochastic | $O(\epsilon^{-2})$ | $T_g$ | $\epsilon$-DG |
| OMD (Mertikopoulos et al., 2018) | coherence | stochastic | N/A | $2T_g$ | asymptotic |
| IPP (Lin et al., 2018) | MVI has solution | stochastic | $O(\epsilon^{-6})$ | $T_g$ | $\epsilon$-SP |
| Alternating Gradient (Gidel et al., 2019) | bilinear form [4] | deterministic | $O(\log(1/\epsilon))$ | $T_g$ | $\epsilon$-optim |
| SVRE (Chavdarova et al., 2019) | strong-monotonicity finite sum | stochastic finite sum | $O(\log(1/\epsilon))$ | $(n + \frac{L}{\mu})T_g$ [5] | $\epsilon$-optim |
| Extragradient (Azizian et al., 2019) | strong-monotonicity | deterministic | $O(\log(1/\epsilon))$ | $2T_g$ | $\epsilon$-optim |
| OSG (this work) | MVI has solution | stochastic | $O(\epsilon^{-4})$ | $T_g$ | $\epsilon$-SP |
| OAdagrad (this work) | MVI has solution | stochastic | $O(\epsilon^{-\frac{2}{1-\alpha}})$ | $T_g$ | $\epsilon$-SP |

Table 1: Summary of different algorithms with IC (Iteration Complexity), PC (Per-iteration Complexity) to find $\epsilon$-SP ($\epsilon$-first-order Stationary Point), $\epsilon$-DG ($\epsilon$-Duality Gap, i.e. a point $(\hat{\mathbf{u}}, \hat{\mathbf{v}})$ such that $\max_{\mathbf{v}} F(\hat{\mathbf{u}}, \mathbf{v}) - \min_{\mathbf{u}} F(\mathbf{u}, \hat{\mathbf{v}}) \leq \epsilon$), or $\epsilon$-optim ($\epsilon$-close to the set of optimal solution). $T_g$ stands for the time complexity for invoking one stochastic first-order oracle.

non-concave min-max problems. Table 1 provides a complete overview of our results and existing results. It is hard to give justice to the large body of work on min-max optimization, so we refer the interested reader to Appendix B that gives a comprehensive survey of related previous methods that are not covered in this Table.

Our main goal is to design stochastic first-order algorithms with **low iteration complexity**, **low per-iteration cost** and suitable for **a general class of non-convex non-concave min-max problems**. The main tool we use in our analysis is variational inequality.

Let $T : \mathbb{R}^d \mapsto \mathbb{R}^d$ be an operator and $\mathcal{X} \subset \mathbb{R}^d$ is a closed convex set. The *Stampacchia Variational Inequality* (SVI) problem (Hartman & Stampacchia, 1966) is defined by the operator $T$ and $\mathcal{X}$ and denoted by $\text{SVI}(T, \mathcal{X})$. It consists of finding $\mathbf{x}_* \in \mathcal{X}$ such that $\langle T(\mathbf{x}_*), \mathbf{x} - \mathbf{x}_* \rangle \geq 0$ for $\forall \mathbf{x} \in \mathcal{X}$. A similar one is *Minty Variational Inequality* (MVI) problem (Minty et al., 1962) denoted by $\text{MVI}(T, \mathcal{X})$, which consists of finding $\mathbf{x}_*$ such that $\langle T(\mathbf{x}), \mathbf{x} - \mathbf{x}_* \rangle \geq 0$ for $\forall \mathbf{x} \in \mathcal{X}$. Min-max optimization is closely related to variational inequalities. The corresponding SVI and MVI for the min-max problem are defined through $T(\mathbf{x}) = [\nabla_{\mathbf{u}} F(\mathbf{u}, \mathbf{v}), -\nabla_{\mathbf{v}} F(\mathbf{u}, \mathbf{v})]^\top$ with $\mathbf{x} = (\mathbf{u}, \mathbf{v})$.

Our main contributions are summarized as follows:

- Following (Daskalakis et al., 2017), we extend optimistic stochastic gradient (OSG) analysis beyond the bilinear and unconstrained case, by assuming the Lipschitz continuity of the operator $T$ and the existence of a solution for the variational inequality $\text{MVI}(T, \mathcal{X})$. These conditions were considered in the analysis of the stochastic extragradient algorithm in (Iusem et al., 2017). We analyze a variant of Optimistic Stochastic Gradient (OSG) under these conditions, inspired by the analysis of (Iusem et al., 2017). We show that OSG achieves state-of-the-art iteration complexity $O(1/\epsilon^4)$ for finding an $\epsilon$-first-order stationary point. Note that our OSG variant only requires invoking one stochastic first-order oracle

---

[3]Note that the pseudo-monotonicity assumption used by (Iusem et al., 2017) can also be replaced by our MVI assumption in their proof. The main difference between our OSG and the stochastic extragradient method in (Iusem et al., 2017) is the number of stochastic gradient calculations in each iteration.

[4]Here the bilinear game is defined as $\min_{\mathbf{u} \in \mathbb{R}^p} \max_{\mathbf{v} \in \mathbb{R}^q} \mathbf{u}^\top A \mathbf{v} + \mathbf{u}^\top \mathbf{a} + \mathbf{b} \mathbf{v}^\top$, where the smallest singular value of $A \in \mathbb{R}^{p \times q}$ is positive, $\mathbf{a} \in \mathbb{R}^{p \times 1}$, $\mathbf{b} \in \mathbb{R}^{1 \times q}$.

[5]Here $n, L, \mu$ denote the number of components in the finite sum structure, Lipschitz constant and strong-monotonicity parameter of the operator of variational inequality respectively.

while enjoying the state-of-the-art iteration complexity achieved by stochastic extragradient method (Iusem et al., 2017).

- Under the same conditions, we design an adaptive gradient algorithm named Optimistic Adagrad (OAdagrad), and show that it enjoys better adaptive complexity $O\left(\epsilon^{-\frac{2}{1-\alpha}}\right)$, where $\alpha$ characterizes the growth rate of cumulative stochastic gradient and $0 \leq \alpha \leq 1/2$. Similar to Adagrad (Duchi et al., 2011), our main innovation is in considering variable metrics according to the geometry of the data in order to achieve potentially faster convergence rate for a class of nonconvex-nonconcave min-max games. Note that this adaptive complexity improves upon the non-adaptive one (i.e., $O(1/\epsilon^4)$) achieved by OSG. To the best of our knowledge, we establish the first known adaptive complexity for adaptive gradient algorithms in a class of non-convex non-concave min-max problems.

- We demonstrate the effectiveness of our algorithms in GAN training on CIFAR10 data. Empirical results identify an important reason behind why adaptive gradient methods behave well in GANs, which is due to the fact that the cumulative stochastic gradient grows in a slow rate. We also show that OAdagrad outperforms Simultaneous Adam in sample quality in ImageNet generation using self-attention GANs (Zhang et al., 2018). This confirms the superiority of OAdagrad in min-max optimization.

## 2 PRELIMINARIES AND NOTATIONS

In this section, we fix some notations and give formal definitions of variational inequalities, and their relationship to the min-max problem (1).

**Notations.** Let $\mathcal{X} \subset \mathbb{R}^d$ be a closed convex set, and $\|\cdot\|$ the euclidean norm. We note $\Pi_{\mathcal{X}}$ the projection operator, i.e. $\Pi_{\mathcal{X}}(\mathbf{y}) = \arg\min_{\mathbf{x} \in \mathcal{X}} \|\mathbf{y} - \mathbf{x}\|^2$. Define $T(\mathbf{x}) = [\nabla_{\mathbf{u}} F(\mathbf{u}, \mathbf{v}), -\nabla_{\mathbf{v}} F(\mathbf{u}, \mathbf{v})]^\top$ with $\mathbf{x} = (\mathbf{u}, \mathbf{v})$ in problem (1). At every point $\mathbf{x} \in \mathcal{X}$, we don't have access to $T(\mathbf{x})$ and have only access to a noisy observations of $T(\mathbf{x})$. That is, $T(\mathbf{x}; \xi)$, where $\xi$ is a random variable with distribution $\mathcal{D}$. For the ease of presentation, we use the terms *stochastic gradient* and *stochastic first-order oracle* interchangeably to stand for $T(\mathbf{x}; \xi)$ in the min-max setting.

**Definition 1** (Monotonicity). *An operator $T$ is monotone if $\langle T(\mathbf{x}) - T(\mathbf{y}), \mathbf{x} - \mathbf{y} \rangle \geq 0$ for $\forall \mathbf{x}, \mathbf{y} \in \mathcal{X}$. An operator $T$ is pseudo-monotone if $\langle T(\mathbf{x}), \mathbf{y} - \mathbf{x} \rangle \geq 0 \Rightarrow \langle T(\mathbf{y}), \mathbf{y} - \mathbf{x} \rangle \geq 0$ for $\forall \mathbf{x}, \mathbf{y} \in \mathcal{X}$. An operator $T$ is $\gamma$-strongly-monotone if $\langle T(\mathbf{x}) - T(\mathbf{y}), \mathbf{x} - \mathbf{y} \rangle \geq \frac{\gamma}{2} \|\mathbf{x} - \mathbf{y}\|^2$ for $\forall \mathbf{x}, \mathbf{y} \in \mathcal{X}$.*

We give here formal definitions of monotonic operators $T$ and the $\epsilon$-first-order stationary point.

**Definition 2** ($\epsilon$-First-Order Stationary Point). *A point $\mathbf{x} \in \mathcal{X}$ is called $\epsilon$-first-order stationary point if $\|T(\mathbf{x})\| \leq \epsilon$.*

**Remark**: We make the following observations:

(a). From the definition, it is evident that strong-monotonicity $\Rightarrow$ monotonicity $\Rightarrow$ pseudo-monotonicity. Assuming SVI has a solution and pseudo-monotonicity of the operator $T$ imply that MVI$(T, \mathcal{X})$ has a solution. To see that, assume that SVI has a nonempty solution set, i.e. there exists $\mathbf{x}_*$ such that $\langle T(\mathbf{x}_*), \mathbf{y} - \mathbf{x}_* \rangle \geq 0$ for any $\mathbf{y}$. Noting that pseudo-monotonicity means that for every $\mathbf{y}, \mathbf{x}$, $\langle T(\mathbf{x}), \mathbf{y} - \mathbf{x} \rangle \geq 0$ implies $\langle T(\mathbf{y}), \mathbf{y} - \mathbf{x} \rangle \geq 0$, we have $\langle T(\mathbf{y}), \mathbf{y} - \mathbf{x}_* \rangle \geq 0$ for any $\mathbf{y}$, which means that $\mathbf{x}_*$ is the solution of Minty variational inequality. Note that the reverse may not be true and an example is provided in Appendix G.

(b). For the min-max problem (1), when $F(\mathbf{u}, \mathbf{v})$ is convex in $\mathbf{u}$ and concave in $\mathbf{v}$, $T$ is monotone. And, therefore solving SVI$(T, \mathcal{X})$ is equivalent to solving (1). When $T$ is not monotone, by assuming $T$ is Lipschitz continuous, it can be shown that the solution set of (1) is a subset of the solution set of SVI$(T, \mathcal{X})$. However, even solving SVI$(T, \mathcal{X})$ is NP-hard in general and hence we resort to finding an $\epsilon$-first-order stationary point.

Throughout the paper, we make the following assumption:

**Assumption 1.**    *(i). $T$ is L-Lipschitz continuous, i.e. $\|T(\mathbf{x}_1) - T(\mathbf{x}_2)\|_2 \leq L\|\mathbf{x}_1 - \mathbf{x}_2\|_2$ for $\forall \mathbf{x}_1, \mathbf{x}_2 \in \mathcal{X}$.*

*(ii). MVI$(T, \mathcal{X})$ has a solution, i.e. there exists $\mathbf{x}_*$ such that $\langle T(\mathbf{x}), \mathbf{x} - \mathbf{x}_* \rangle \geq 0$ for $\forall \mathbf{x} \in \mathcal{X}$.*

*(iii). For $\forall \mathbf{x} \in \mathcal{X}$, $\mathbb{E}\left[T(\mathbf{x}; \xi)\right] = T(\mathbf{x})$, $\mathbb{E}\left\|T(\mathbf{x}; \xi) - T(\mathbf{x})\right\|^2 \leq \sigma^2$.*

**Remark:** Assumptions (i) and (iii) are commonly used assumptions in the literature of variational inequalities and non-convex optimization (Juditsky et al., 2011; Ghadimi & Lan, 2013; Iusem et al., 2017). Assumption (ii) is used frequently in previous work focusing on analyzing algorithms that solve non-monotone variational inequalities (Iusem et al., 2017; Lin et al., 2018; Mertikopoulos et al., 2018). Assumption (ii) is weaker than other assumptions usually considered, such as pseudo-monotonicity, monotonicity, or coherence as assumed in (Mertikopoulos et al., 2018). For non-convex minimization problem, it has been shown that this assumption holds while using SGD to learn neural networks (Li & Yuan, 2017; Kleinberg et al., 2018; Zhou et al., 2019).

## 3 OPTIMISTIC STOCHASTIC GRADIENT

This section serves as a warm-up and motivation of our main theoretical contribution presented in the next section. Inspired by (Iusem et al., 2017), we present an algorithm called Optimistic Stochastic Gradient (OSG) that saves the cost of the additional oracle call as required in (Iusem et al., 2017) and maintains the same iteration complexity. The main algorithm is described in Algorithm 1, where $m_t$ denotes the minibatch size for estimating the first-order oracle. It is worth mentioning that Algorithm 1 becomes stochastic extragradient method if one changes $T(\mathbf{z}_{k-1}; \xi_{k-1}^i)$ to $T(\mathbf{x}_{k-1}; \xi_{k-1}^i)$ in line 3. Stochastic extragradient method requires to compute stochastic gradient over both sequences $\{\mathbf{x}_k\}$ and $\{\mathbf{z}_k\}$. In contrast, $\{\mathbf{x}_k\}$ is an ancillary sequence in OSG and the stochastic gradient is only computed over the sequence of $\{\mathbf{z}_k\}$. Thus, stochastic extragradient method is twice as expensive as OSG in each iteration. In some tasks (e.g. training GANs) where the stochastic gradient computation is expensive, OSG is numerically more appealing.

---

**Algorithm 1** Optimistic Stochastic Gradient (OSG)

---

1: **Input:** $\mathbf{z}_0 = \mathbf{x}_0 = 0$
2: **for** $k = 1, \ldots, N$ **do**
3: $\quad \mathbf{z}_k = \Pi_{\mathcal{X}} \left[ \mathbf{x}_{k-1} - \eta \cdot \frac{1}{m_{k-1}} \sum_{i=1}^{m_{k-1}} T(\mathbf{z}_{k-1}; \xi_{k-1}^i) \right]$
4: $\quad \mathbf{x}_k = \Pi_{\mathcal{X}} \left[ \mathbf{x}_{k-1} - \eta \cdot \frac{1}{m_k} \sum_{i=1}^{m_k} T(\mathbf{z}_k; \xi_k^i) \right]$
5: **end for**

---

**Remark:** When $\mathcal{X} = \mathbb{R}^d$, the update in Algorithm 1 becomes the algorithm in (Daskalakis et al., 2017), i.e.

$$\mathbf{z}_{k+1} = \mathbf{z}_k - 2\eta \cdot \frac{1}{m_{k-1}} \sum_{i=1}^{m_k} T(\mathbf{z}_k; \xi_k^i) + \eta \cdot \frac{1}{m_{k-1}} \sum_{i=1}^{m_{k-1}} T(\mathbf{z}_{k-1}; \xi_{k-1}^i) \tag{2}$$

The detailed derivation of (2) can be found in Appendix F.

**Theorem 1.** *Suppose that Assumption 1 holds. Let $r_\alpha(\mathbf{z}_k) = \|\mathbf{z}_k - \Pi_{\mathcal{X}}(\mathbf{z}_k - \alpha T(\mathbf{z}_k))\|$. Let $\eta \leq 1/9L$ and run Algorithm 1 for $N$ iterations. Then we have*

$$\frac{1}{N} \sum_{k=1}^{N} \mathbb{E}\left[r_\eta^2(\mathbf{z}_k)\right] \leq \frac{8\|\mathbf{x}_0 - \mathbf{x}_*\|^2}{N} + \frac{100\eta^2}{N} \sum_{k=0}^{N} \frac{\sigma^2}{m_k},$$

**Corollary 1.** *Consider the unconstrained case where $\mathcal{X} = \mathbb{R}^d$. Let $\eta \leq 1/9L$, and we have*

$$\frac{1}{N} \sum_{k=1}^{N} \mathbb{E}\|T(\mathbf{z}_k)\|_2^2 \leq \frac{8\|\mathbf{x}_0 - \mathbf{x}_*\|^2}{\eta^2 N} + \frac{100}{N} \sum_{k=0}^{N} \frac{\sigma^2}{m_k}, \tag{3}$$

**Remark:** There are two implications of Corollary 1.

- (Increasing Minibatch Size) Let $\eta = \frac{1}{9L}$, $m_k = k + 1$. To guarantee $\frac{1}{N}\sum_{k=1}^{N}\mathbb{E}\|T(\mathbf{z}_k)\|_2^2 \leq \epsilon^2$, the total number of iterations is $N = \widetilde{O}(\epsilon^{-2})$, and the total complexity is $\sum_{k=1}^{N} m_k = \widetilde{O}(\epsilon^{-4})$, where $\widetilde{O}(\cdot)$ hides a logarithmic factor of $\epsilon$.

- (Constant Minibatch Size) Let $\eta = \frac{1}{9L}$, $m_k = 1/\epsilon^2$. To guarantee $\frac{1}{N}\sum_{k=1}^{N}\mathbb{E}\|T(\mathbf{z}_k)\|_2^2 \leq \epsilon^2$, the total number of iterations is $N = O(\epsilon^{-2})$, and the total complexity is $\sum_{k=0}^{N} m_k = O(\epsilon^{-4})$.

# 4 OPTIMISTIC ADAGRAD

## 4.1 ADAGRAD FOR MINIMIZATION PROBLEMS

Before introducing Optimistic Adagrad, we present here a quick overview of Adagrad (Duchi et al., 2011). The main objective in Adagrad is to solve the following minimization problem:

$$\min_{\mathbf{w}\in\mathbb{R}^d} F(\mathbf{w}) = \mathbb{E}_{\zeta\sim\mathcal{P}}f(\mathbf{w};\zeta) \tag{4}$$

where $\mathbf{w}$ is the model parameter, and $\zeta$ is an random variable following distribution $\mathcal{P}$. The update rule of Adagrad is

$$\mathbf{w}_{t+1} = \mathbf{w}_t - \eta H_t^{-1}\hat{\mathbf{g}}_t, \tag{5}$$

where $\eta > 0$, $\hat{\mathbf{g}}_t = \nabla f(\mathbf{w}_t;\zeta_t)$, $H_t = \text{diag}\left(\left(\sum_{i=1}^{t}\hat{\mathbf{g}}_i \circ \hat{\mathbf{g}}_i\right)^{\frac{1}{2}}\right)$ with $\circ$ denoting the Hadamard product. Adagrad when taking $H_t = I$ reduces to SGD. Different from SGD, Adagrad dynamically incorporates knowledge of history gradients to perform more informative gradient-based learning. When solving a convex minimization problem and the gradient is sparse, Adagrad converges faster than SGD. There are several variants of Adagrad, including Adam (Kingma & Ba, 2014), RM-SProp (Tieleman & Hinton, 2012), and AmsGrad (Reddi et al., 2019). All of them share the spirit, as they take advantage of the information provided by the history of gradients. Wilson et al. (2017) provide a complete overview of different adaptive gradient methods in a unified framework. It is worth mentioning that Adagrad can not be directly applied to solve non-convex non-concave min-max problems with provable guarantee.

## 4.2 OPTIMISTIC ADAGRAD FOR MIN-MAX OPTIMIZATION

Our second algorithm named Optimistic Adagrad (OAdagrad) is an adaptive variant of OSG, which also updates minimization variable and maximization variable simultaneously. The key difference between OSG and OAdagrad is that OAdagrad inherits ideas from Adagrad to construct variable metric based on history gradients information, while OSG only utilizes a fixed metric. This difference helps us establish faster adaptive convergence under some mild assumptions. Note that in OAdagrad we only consider the unconstrained case, i.e. $\mathcal{X} = \mathbb{R}^d$.

**Assumption 2.** *(i). There exists $G > 0$ and $\delta > 0$ such that $\|T(\mathbf{z};\xi)\|_2 \leq G$, $\|T(\mathbf{z};\xi)\|_\infty \leq \delta$ for all $\mathbf{z}$ almost surely.*

*(ii). There exists a universal constant $D > 0$ such that $\|\mathbf{x}_k\|_2 \leq D/2$ for $k = 1,\ldots,N$, and $\|\mathbf{x}_*\|_2 \leq D/2$.*

**Remark**: Assumption 2 (i) is a standard one often made in literature (Duchi et al., 2011). Assumption 2 (ii) holds when we use normalization layers in the discriminator and generator such as spectral normalization of weights (Miyato et al., 2018; Zhang et al., 2018), that will keep the norms of the weights bounded. Regularization techniques such as weight decay also ensure that the weights of the networks remain bounded throughout the training.

Define $\hat{\mathbf{g}}_k = \frac{1}{m}\sum_{i=1}^{m}T(\mathbf{z}_k;\xi_k^i)$, $\|\mathbf{x}\|_H = \sqrt{\langle\mathbf{x}, H\mathbf{x}\rangle}$. Denote $\hat{\mathbf{g}}_{0:k}$ by the concatenation of $\hat{\mathbf{g}}_0,\ldots,\hat{\mathbf{g}}_k$, and denote $\hat{\mathbf{g}}_{0:k,i}$ by the $i$-th row of $\hat{\mathbf{g}}_{0:k}$.

---

**Algorithm 2** Optimistic AdaGrad (OAdagrad)

---

1: **Input:** $\mathbf{z}_0 = \mathbf{x}_0 = 0$, $H_0 = \delta I$
2: **for** $k = 1, \ldots, N$ **do**
3:     $\mathbf{z}_k = \mathbf{x}_{k-1} - \eta H_{k-1}^{-1} \widehat{\mathbf{g}}_{k-1}$
4:     $\mathbf{x}_k = \mathbf{x}_{k-1} - \eta H_{k-1}^{-1} \widehat{\mathbf{g}}_k$
5:     Update $\widehat{\mathbf{g}}_{0:k} = [\widehat{\mathbf{g}}_{0:k-1} \ \widehat{\mathbf{g}}_k]$, $s_{k,i} = \|\widehat{\mathbf{g}}_{0:k,i}\|$, $i = 1, \ldots, d$ and set $H_k = \delta I + \text{diag}(s_{k-1})$
6: **end for**

---

**Theorem 2.** *Suppose Assumption 1 and 2 hold. Suppose $\|\widehat{\mathbf{g}}_{1:k,i}\|_2 \leq \delta k^\alpha$ with $0 \leq \alpha \leq 1/2$ for every $i = 1, \ldots, d$ and every $k = 1, \ldots, N$. When $\eta \leq \frac{\delta}{9L}$, after running Algorithm 2 for N iterations, we have*

$$\frac{1}{N} \sum_{k=1}^{N} \mathbb{E}\|T(\mathbf{z}_k)\|_{H_{k-1}^{-1}}^2 \leq \frac{8D^2\delta^2(1 + d(N-1)^\alpha)}{\eta^2 N} + \frac{100 \left( \sigma^2/m + d \left( 2\delta^2 N^\alpha + G^2 \right) \right)}{N}. \quad (6)$$

*To make sure $\frac{1}{N} \sum_{k=1}^{N} \mathbb{E}\|T(\mathbf{z}_k)\|_{H_{k-1}^{-1}}^2 \leq \epsilon^2$, the number of iterations is $N = O\left( \epsilon^{-\frac{2}{1-\alpha}} \right)$.*

**Remark**:

- Note that the convergence measure used in Theorem 2 is different from that in Corollary 1. However we show that under the measure used in Theorem 2, OSG (Algorithm 1) still has complexity $O(1/\epsilon^4)$. By the construction of $H_k$ in Algorithm 2, we know that $\|T(\mathbf{z})\|_{H_k^{-1}} \leq \|T(\mathbf{z})\|_{H_0^{-1}}$ for any $k \geq 0$ and any $\mathbf{z}$, and hence $\frac{1}{N} \sum_{k=1}^{N} \mathbb{E}\|T(\mathbf{z}_k)\|_{H_{k-1}^{-1}}^2 \leq \frac{1}{N} \sum_{k=1}^{N} \mathbb{E}\|T(\mathbf{z}_k)\|_{H_0^{-1}}^2 = \frac{1}{\delta} \cdot \frac{1}{N} \sum_{k=1}^{N} \mathbb{E}\|T(\mathbf{z}_k)\|_2^2$. By Corollary 1, we know that OSG still requires $O(1/\epsilon^4)$ complexity to guarantee that $\frac{1}{N} \sum_{k=1}^{N} \mathbb{E}\|T(\mathbf{z}_k)\|_{H_{k-1}^{-1}}^2 \leq \epsilon^2$.

- We denote $\widehat{\mathbf{g}}_{1:k}$ by the cumulative stochastic gradient, where $\|\widehat{\mathbf{g}}_{1:k,i}\|_2 \leq \delta k^\alpha$ characterizes the growth rate of the gradient in terms of $i$-th coordinate. In our proof, a key quantity is $\sum_{i=1}^{d} \|\widehat{\mathbf{g}}_{1:k,i}\|_2$ that crucially affects the computational complexity of Algorithm 2. Since $\sum_{i=1}^{d} \|\widehat{\mathbf{g}}_{1:k,i}\|_2 \leq \delta d k^\alpha$, in the worst case, $\alpha = \frac{1}{2}$. But in practice, the stochastic gradient is usually sparse, and hence $\alpha$ can be strictly smaller than $\frac{1}{2}$.

- As shown in Theorem 2, the minibatch size used in Algorithm 2 for estimating the first-order oracle can be any positive constant and independent of $\epsilon$. This is more practical than the results established in Theorem 1, since the minibatch size in Theorem 1 does either increase in terms of number of iterations or is dependent on $\epsilon$. When $\alpha = \frac{1}{2}$, the complexity of Algorithm 2 is $O(1/\epsilon^4)$, which matches the complexity stated in Theorem 1. When $\alpha < \frac{1}{2}$, the complexity of OAdagrad given in Algorithm 2 is $O\left( \epsilon^{-\frac{2}{1-\alpha}} \right)$, i.e., strictly better than that of OSG given in Algorithm 1.

**Comparison with Alternating Adam and Optimistic Adam**    Alternating Adam is very popular in GAN training (Goodfellow et al., 2014; Arjovsky et al., 2017; Gulrajani et al., 2017; Brock et al., 2018). In Alternating Adam, one alternates between multiple steps of Adam on the discriminator and a single step of Adam on the generator. The key difference between OAdagrad and Alternating Adam is that OAdagrad updates the discriminator and generator simultaneously. It is worth mentioning that OAdagrad naturally fits into the framework of Optimistic Adam proposed in (Daskalakis et al., 2017). Taking $\beta_1 = 0, \beta_2 \to 1$ in their Algorithm 1 reduces to OAdagrad with annealing learning rate. To the best of our knowledge, there is no convergence proof for Alternating Adam for non-convex non-concave problems. Our convergence proof for OAdagrad provides a theoretical justification of a special case of Optimistic Adam.

## 5 EXPERIMENTS

**WGAN-GP on CIFAR10**    In the first experiment, we verify the effectiveness of the proposed algorithms in GAN training using the PyTorch framework (Paszke et al., 2017). We use Wasserstein

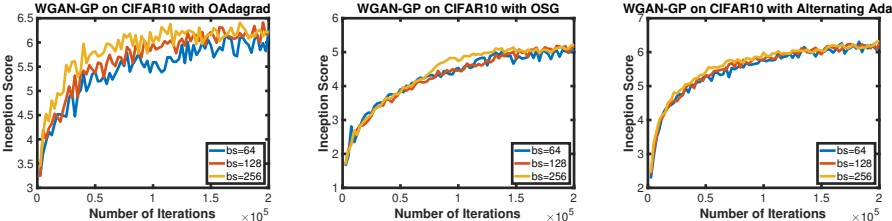

Figure 1: OAdagrad, OSG and Alternating Adam for WGAN-GP on CIFAR10 data

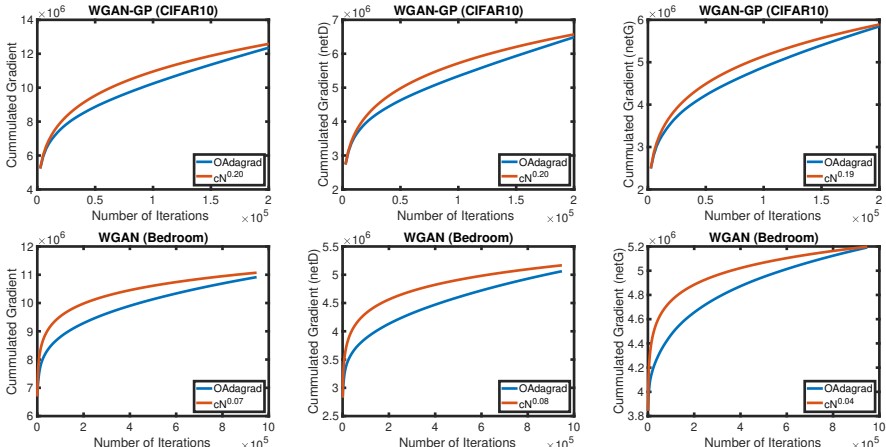

Figure 2: Cumulative Stochastic Gradient as a function of number of iterations, where netD and netG stand for the discriminator and generator respectively. The blue curve and red curve stand for the growth rate of the cummulative stochastic gradient for OAdagrad and its corresponding tightest polynomial growth upper bound, respectively.

GAN with gradient penalty (WGAN-GP) (Gulrajani et al., 2017) and CIFAR10 data in our experiments. The architectures of discriminator and generator, and the penalty parameter in WGAN-GP are set to be same as in the original paper. We compare Alternating Adam, OSG and OAdagrad, where the Alternating Adam is to run 5 steps of Adam on the discriminator before performing 1 step of Adam on the generator. We try different batch sizes $(64, 128, 256)$ for each algorithm. For each algorithm, we tune the learning rate in the range of $\{1 \times 10^{-3}, 2 \times 10^{-4}, 1 \times 10^{-4}, 2 \times 10^{-5}, 1 \times 10^{-5}\}$ when using batch size 64, and use the same learning rate for batch size 128 and 256. We report Inception Score (IS) (Salimans et al., 2016) as a function of number of iterations. Figure 1 suggests that OAdagrad performs better than OSG and Alternating Adam, and OAdagrad results in higher IS. We compare the generated CIFAR10 images associated with these three methods, which is included in Appendix A. We also provide experimental results to compare the performance of different algorithms using different minibatch sizes, which are included in Appendix E.

**Growth Rate of Cumulative Stochastic Gradient** In the second experiment, we employ OAdagrad to train GANs and study the growth rate of the cumulative stochastic gradient (i.e., $\sum_{i=1}^{d} \|\widehat{\mathbf{g}}_{1:N,i}\|_2$). We tune the learning rate from $\{1 \times 10^{-3}, 2 \times 10^{-4}, 1 \times 10^{-4}, 2 \times 10^{-5}, 1 \times 10^{-5}\}$ and choose batch size to be 64. In Figure 2, the blue curve and red curve stand for the growth rate for OAdagrad and its corresponding tightest polynomial growth upper bound respectively. $N$ is the number of iterations, and $c$ is a multiplicative constant such that the red curve and blue curve overlaps at the starting point of the training. The degree of the polynomial is determined using binary search. We can see that the growth rate of cumulative stochastic gradient grows very slowly in GANs (the worst-case polynomial degree is 0.5, but it is 0.2 for WGAN-GP on CIFAR10 and 0.07 for WGAN on LSUN Bedroom dataset). As predicted by our theory, this behavior explains the faster convergence of OAdagrad versus OSG, consistent with what is observed empirically in Figure 1.

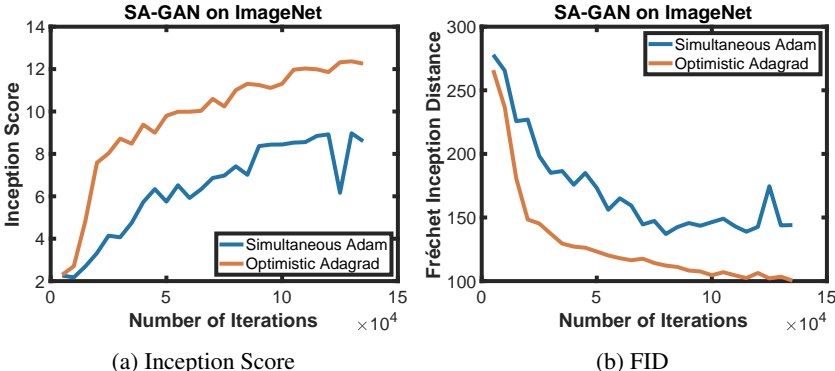

(a) Inception Score        (b) FID

Figure 3: Self-Attention GAN on ImageNet, with evaluation using Official TensorFlow Inception Score and Official TensorFlow FID. We see that OAdagard indeed outperforms Simultaneous Adam in terms of the (TensorFlow) Inception score (higher is better), and in terms of (TensorFlow) Fréchet Inception Distance (lower is better). We don't report here Alternating Adam since in our run it has collapsed.

**Self-Attention GAN on ImageNet** In the third experiment, we consider GAN training on large-scale dataset. We use the model from Self-Attention GAN (Zhang et al., 2018) (SA-GAN) and ImageNet as our dataset. Note that in this setting the boundedness of both generator ($G$) and discriminator ($D$) is ensured by spectral normalization of both $G$ and $D$. Three separate experiments are performed, including Alternating Adam (baseline), Simultaneous Adam (Mescheder et al., 2017), and OAdagrad. It should be mentioned that the update rule of Simultaneous Adam involves performing Adam-type update for discriminator and generator simultaneously. Training is performed with batch size 128 for all experiments.

For the baseline experiment (Alternating Adam) we use the default settings and hyper parameters reported in SA-GAN (Zhang et al., 2018) (note that we are not using the same batch size of 256 as in (Zhang et al., 2018) due to limited computational resources). In our experience, Alternating Adam training for a batch size of 128 with same learning rate as in SA-GAN (0.0001 for generator and 0.0004 for discriminator) collapsed. This does not mean that Alternating Adam fails, it just needs more tuning to find the correct range of learning rates for the particular batch size we have. With the hyperparameters ranges we tried Alternating Adam collapsed, with extra tuning efforts and an expensive computational budget Alternating Adam would eventually succeed. This is inline with the large scale study in (Lucic et al., 2018) that states that given a large computational budget for tuning hyper-parameters most GANs training succeed equally.

For both OAdagrad and Simultaneous Adam, we use different learning rate for generator and discriminator, as suggested in (Heusel et al., 2017). Specifically, the learning rates used are $10^{-3}$ for the generator and $4 \times 10^{-5}$ for the discriminator. We report both Inception Score (IS) and Fréchet Inception Distance (Heusel et al., 2017) (FID) as a function of number of iterations.

We compare the generated ImageNet images associated with the three optimization methods in Appendix A. Since Alternating Adam collapsed we don't report its Inception Score or FID. As it can be seen in Figure 3 and Appendix A, OAdagrad outperforms simultaneous Adam in quantitative metrics (IS and FID) and in sample quality generation. Future work will include investigating whether OAdagrad would benefit from training with larger batch size, in order to achieve state-of-the-art results.

## 6 CONCLUSION

In this paper, we explain the effectiveness of adaptive gradient methods in training GANs from both theoretical and empirical perspectives. Theoretically, we provide two efficient stochastic algorithms for solving a class of min-max non-convex non-concave problems with state-of-the-art computational complexities. We also establish adaptive complexity results for an Adagrad-style algorithm by using coordinate-wise stepsize according to the geometry of the history data. The algorithm is proven to enjoy faster adaptive convergence than its non-adaptive counterpart when the gradient is

sparse, which is similar to Adagrad applied to convex minimization problem. We have conducted extensive empirical studies to verify our theoretical findings. In addition, our experimental results suggest that the reason why adaptive gradient methods deliver good practical performance for GAN training is due to the slow growth rate of the cumulative stochastic gradient.

## ACKNOWLEDGMENTS

The authors thank the anonymous reviewers for their helpful comments. M. Liu and T. Yang are partially supported by National Science Foundation CAREER Award 1844403. M. Liu would like to thank Xiufan Yu from Pennsylvania State University and Zehao Dou from Yale University for helpful discussions.

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

## A    MORE EXPERIMENTAL RESULTS

**Comparison of Generated CIFAR10 Images by Different Optimization Methods** In this section, we report the generated CIFAR10 images during the training of WGAN-GP by three optimization methods (OSG, OAdagrad, Alternating Adam). Every method uses batch size 64, and 1 iteration represents calculating the stochastic gradient with minibatch size 64 once. Figure 4 consists of images by three optimization methods at iteration 8000. Visually we can see that OAdagrad is better than Alternating Adam, and both of them are significantly better than OSG. It is consistent with the inception score results reported in Figure 1, and it also illustrates the tremendous benefits delivered by adaptive gradient methods when training GANs.

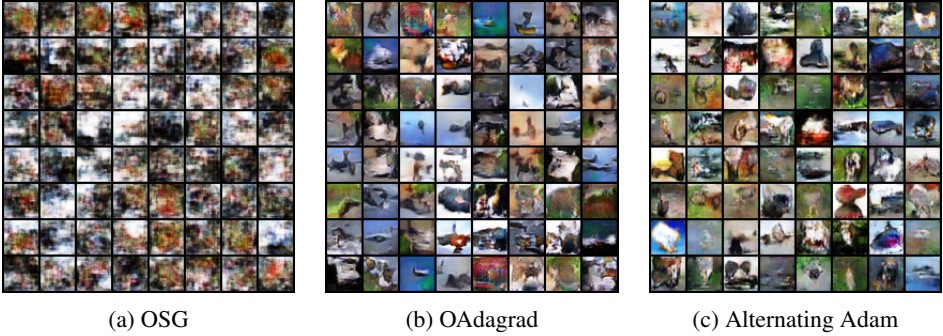

(a) OSG                    (b) OAdagrad                    (c) Alternating Adam

Figure 4: WGAN-GP: Generated CIFAR10 images using different optimization methods at iteration 8000.

**Comparison of Generated ImageNet Images by Different Optimization Methods** In this section, we report the generated ImageNet images during the training of Self-Attention GAN by three optimization methods (OAdagrad, Simultaneous Adam, Alternating Adam). Every method uses batch size 128 and 1 iteration represents calculating the stochastic gradient with minibatch 128 once. Figure 5 consists of images by three optimization methods at iteration 135000. Visually it is apparent that OAdagrad is better than Simultaneous Adam, and both of them are significantly than Alternating Adam.

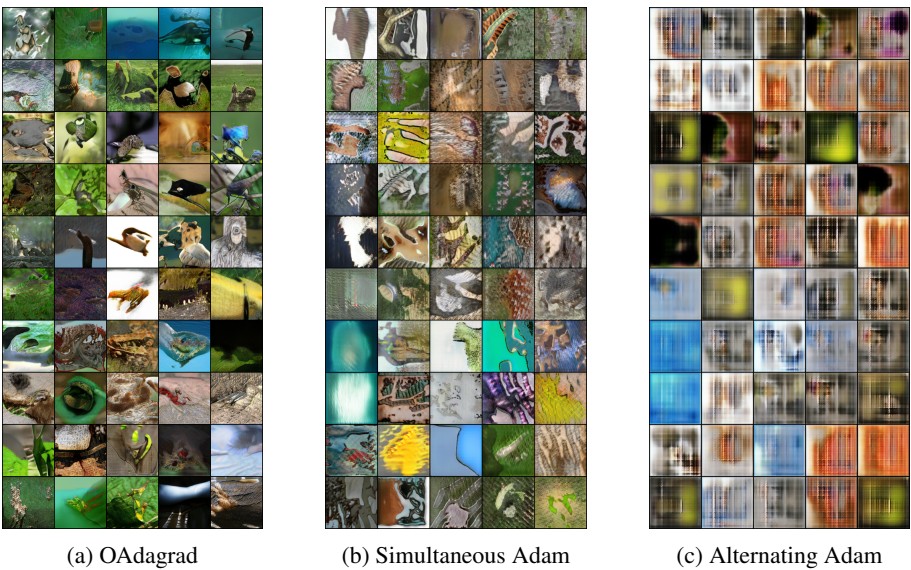

(a) OAdagrad                    (b) Simultaneous Adam                    (c) Alternating Adam

Figure 5: Self-Attention GAN (SA-GAN): Generated ImageNet images using different optimization methods at iteration 135000. OAdagrad produces better quality images than simultaneous Adam. For both Oadagrad and simultaneous Adam we use the same learning rates: 0.001 for generator and 0.00004 for the discriminator. Alternating Adam in our experience with same learning rate as in SA-GAN 0.0001 for generator and 0.0004 for discriminator collapsed. Note that our setting is different from SA-GAN since our batchsize is 128 while it is 256 in SA-GAN. It was also noted in SA-GAN that alternating Adam is hard to train.

**Unofficial PyTorch Inception Score and FID results for SA-GAN on ImageNet**

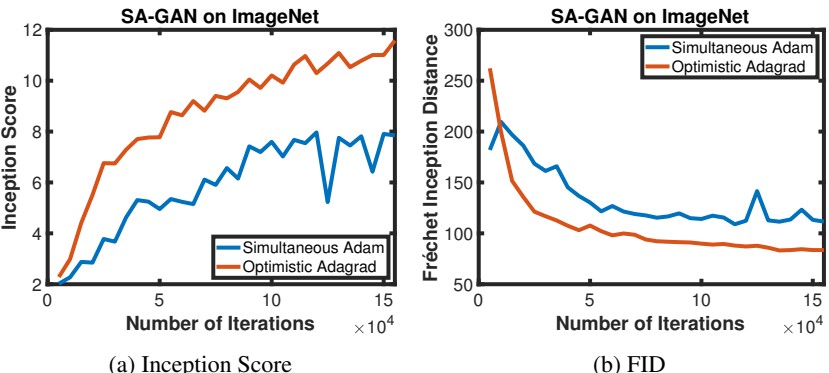

(a) Inception Score          (b) FID

Figure 6: Self-Attention GAN on ImageNet, with evaluation using Unoffical PyTorch Inception Score and Unoffical Pytorch FID. We see that OAdagard indeed outperforms Simultaneous Adam in terms of the (PyTorch) Inception score (higher is better), and in terms of (PyTorch) Fréchet Inception Distance (lower is better). We don't report here Alternating Adam since in our run it has collapsed.

## B    RELATED WORK

**Min-max Optimization and GAN Training**    For convex-concave min-max optimization, the extragradient method was first proposed by (Korpelevich, 1976). Later on, under gradient Lipschitz condition, Nemirovski (2004) extended the idea of extragradient to mirror-prox and obtained the $O(1/N)$ convergence rate in terms of the duality gap (see also (Nesterov, 2007)), where $N$ is the number of iterations. When only the stochastic first-order oracle is available, the stochastic mirror-prox was analyzed by (Juditsky et al., 2011). The convergence rates for both deterministic and stochastic mirror-prox are optimal (Nemirovsky & Yudin, 1983). Recently, Zhao (2019) developed a nearly-optimal stochastic first-order algorithm when the primal variable is strongly convex in the primal variable. Bach & Levy (2019) proposed a universal algorithm that is adaptive to smoothness and noise, and simultaneously achieves optimal convergence rate.

There is a plethora of work analyzing one-sided nonconvex min-max problem, where the objective function is nonconvex in the minimization variable but concave in maximization variable. When the function is weakly-convex in terms of the minimization variable, Rafique et al. (2018) propose a stage-wise stochastic algorithm that approximately solves a convex-concave subproblem by adding a quadratic regularizer and show the first-order convergence of the equivalent minimization problem. Under the same setting, Lu et al. (2019) utilize block-based optimization strategy and show the convergence of the stationarity gap. By further assuming that the function is smooth in the minimization variable, Lin et al. (2019) show that (stochastic) gradient descent ascent is able to converge to the first-order stationary point of the equivalent minimization problem. Liu et al. (2020) cast the problem of stochastic AUC maximization with deep neural networks into a nonconvex-concave min-max problem, show the PL (Polyak-Łojasiewicz) condition holds for the objective of the outer minimization problem, and propose an algorithm and establish its fast convergence rate.

A more challenging problem is the non-convex non-concave min-max problem. Dang & Lan (2015) demonstrate that the deterministic extragradient method is able to converge to $\epsilon$-first-order stationary point with non-asymptotic guarantee. Under the condition that the objective function is weakly-convex and weakly-concave, Lin et al. (2018) designs a stage-wise algorithm, where in each stage a strongly-convex strongly-concave subproblem is constructed by adding quadratic terms and appropriate stochastic algorithms can be employed to approximately solve it. They also show the convergence to the stationary point. Sanjabi et al. (2018) design an alternating deterministic optimization algorithm, in which multiple steps of gradient ascent for dual variable are conducted before one step of gradient descent for primal variable is performed. They show the convergence to stationary point based on the assumption that the inner maximization problem satisfies PL condition (Polyak, 1969). Our work is different from these previous methods in many aspects. In comparison to (Lin et al., 2018), our result does not need the bounded domain assumption. Furthermore, our iteration complexity is $O(1/\epsilon^4)$ to achieve $\epsilon$-first-order stationary point while the corresponding complexity

in (Lin et al., 2018) is $O(1/\epsilon^6)$. When comparing to (Sanjabi et al., 2018), we do not assume that the PL (Polyak-Łojasiewicz) condition holds. Additionally, our algorithm is stochastic and not restricted to the deterministic case. Apparently the most related work to the present one is (Iusem et al., 2017). The stochastic extragradient method analyzed in (Iusem et al., 2017) requires calculation of two stochastic gradients per iteration, while the present algorithm only needs one since it memorizes the stochastic gradient in the previous iteration to guide the update in the current iteration. Nevertheless, we achieve the same iteration complexity as in (Iusem et al., 2017).

There are a body of work analyzing the convergence behavior of min-max optimization algorithms and its application in training GANs (Heusel et al., 2017; Daskalakis & Panageas, 2018; Nagarajan & Kolter, 2017; Grnarova et al., 2017; Yadav et al., 2017; Gidel et al., 2018; Mertikopoulos et al., 2018; Mazumdar et al., 2019). A few of them (Heusel et al., 2017; Daskalakis & Panageas, 2018; Mazumdar et al., 2019) only have asymptotic convergence. Others (Nagarajan & Kolter, 2017; Grnarova et al., 2017; Daskalakis et al., 2017; Yadav et al., 2017; Gidel et al., 2018; Mertikopoulos et al., 2018) focus on more restricted settings. For example, Nagarajan & Kolter (2017); Grnarova et al. (2017) require the concavity of the objective function in terms of dual variable. Yadav et al. (2017); Gidel et al. (2018) assume the objective to be convex-concave. Mertikopoulos et al. (2018) imposes the so-called coherence condition which is stronger than our assumption. Daskalakis et al. (2017) analyze the last-iteration convergence for bilinear problem. Recently, Gidel et al. (2019) analyze the benefits of using negative momentum in alternating gradient descent to improve the training of a bilinear game. Chavdarova et al. (2019) develop a variance-reduced extragradient method and shows its linear convergence under strong monotonicity and finite-sum structure assumptions. Azizian et al. (2019) provide a unified analysis of extragradient for bilinear game, strongly monotone case, and their intermediate cases. However, none of them give non-asymptotic convergence results for the class of non-convex non-concave min-max problem considered in our paper.

## C  PROOF OF THEOREM 1

### C.1  FACTS

Suppose $\mathcal{X} \subset \mathbb{R}^d$ is closed and convex set, then we have

**Fact 1.** *For all $\mathbf{x} \in \mathbb{R}^d$ and $\mathbf{y} \in \mathcal{X}$, $\|\Pi_{\mathcal{X}}(\mathbf{x}) - \mathbf{y}\|^2 + \|\Pi_{\mathcal{X}}(\mathbf{x}) - \mathbf{x}\|^2 \leq \|\mathbf{x} - \mathbf{y}\|^2$.*

**Fact 2.** *For all $\mathbf{x} \in \mathbb{R}^d$ and $\mathbf{y} \in \mathcal{X}$, $\langle \mathbf{x} - \Pi_{\mathcal{X}}(\mathbf{x}), \mathbf{y} - \Pi_{\mathcal{X}}(\mathbf{x}) \rangle \leq 0$.*

### C.2  LEMMAS

**Lemma 1.** *For $\eta \leq \frac{1}{9L}$, we have*

$$\frac{1}{2} \sum_{k=1}^{N} \|\mathbf{x}_{k-1} - \mathbf{z}_k\|^2 + \frac{1}{2} \sum_{k=1}^{N} \|\mathbf{x}_k - \mathbf{z}_k\|^2 \leq \|\mathbf{x}_0 - \mathbf{x}_*\|^2 - \|\mathbf{x}_N - \mathbf{x}_*\|^2 + 12\eta^2 \sum_{k=0}^{N} \|\epsilon_k\|^2 + \sum_{k=1}^{N} \Lambda_k \quad (7)$$

*Proof.* Let $\mathbf{x}_* \in \mathcal{X}^*$, where $\mathcal{X}^*$ is the set of optimal solutions of $\mathrm{MVI}(T, \mathcal{X})$, i.e. $\langle T(\mathbf{x}), \mathbf{x} - \mathbf{x}_* \rangle \geq 0$ holds for $\forall \mathbf{x} \in \mathcal{X}$. Define $\epsilon_k = \frac{1}{m_k} \sum_{i=1}^{m_k} T(\mathbf{z}_k, \xi_k^i) - T(\mathbf{z}_k)$, and $\widehat{T}(\epsilon_k, \mathbf{z}_k) = T(\mathbf{z}_k) + \epsilon_k$. For

any $\mathbf{x} \in \mathcal{X}$, we have

$$\|\mathbf{x}_k - \mathbf{x}\|^2 = \|\Pi_{\mathcal{X}}\left(\mathbf{x}_{k-1} - \eta\widehat{T}(\epsilon_k, \mathbf{z}_k)\right) - \mathbf{x}\|^2$$

$$\overset{(a)}{\leq} \left\|\mathbf{x}_{k-1} - \eta\widehat{T}(\epsilon_k, \mathbf{z}_k) - \mathbf{x}\right\|^2 - \left\|\mathbf{x}_{k-1} - \eta\widehat{T}(\epsilon_k, \mathbf{z}_k) - \Pi_{\mathcal{X}}\left(\mathbf{x}_{k-1} - \eta\widehat{T}(\epsilon_k, \mathbf{z}_k)\right)\right\|^2$$

$$= \left\|\mathbf{x}_{k-1} - \eta\widehat{T}(\epsilon_k, \mathbf{z}_k) - \mathbf{x}\right\|^2 - \left\|\mathbf{x}_{k-1} - \eta\widehat{T}(\epsilon_k, \mathbf{z}_k) - \mathbf{x}_k\right\|^2$$

$$= \|\mathbf{x}_{k-1} - \mathbf{x}\|^2 - \|\mathbf{x}_{k-1} - \mathbf{x}_k\|^2 + 2\left\langle \mathbf{x} - \mathbf{x}_k, \eta\widehat{T}(\epsilon_k, \mathbf{z}_k)\right\rangle$$

$$= \|\mathbf{x}_{k-1} - \mathbf{x}\|^2 - \|\mathbf{x}_{k-1} - \mathbf{x}_k\|^2 + 2\left\langle \mathbf{x} - \mathbf{z}_k, \eta\widehat{T}(\epsilon_k, \mathbf{z}_k)\right\rangle + 2\left\langle \mathbf{z}_k - \mathbf{x}_k, \eta\widehat{T}(\epsilon_k, \mathbf{z}_k)\right\rangle$$

$$= \|\mathbf{x}_{k-1} - \mathbf{x}\|^2 - \|\mathbf{x}_{k-1} - \mathbf{z}_k + \mathbf{z}_k - \mathbf{x}_k\|^2 + 2\left\langle \mathbf{x} - \mathbf{z}_k, \eta\widehat{T}(\epsilon_k, \mathbf{z}_k)\right\rangle + 2\left\langle \mathbf{z}_k - \mathbf{x}_k, \eta\widehat{T}(\epsilon_k, \mathbf{z}_k)\right\rangle$$

$$= \|\mathbf{x}_{k-1} - \mathbf{x}\|^2 - \|\mathbf{x}_{k-1} - \mathbf{z}_k\|^2 - \|\mathbf{z}_k - \mathbf{x}_k\|^2 - 2\left\langle \mathbf{x}_{k-1} - \mathbf{z}_k, \mathbf{z}_k - \mathbf{x}_k\right\rangle +$$
$$\quad 2\left\langle \mathbf{x} - \mathbf{z}_k, \eta\widehat{T}(\epsilon_k, \mathbf{z}_k)\right\rangle + 2\left\langle \mathbf{z}_k - \mathbf{x}_k, \eta\widehat{T}(\epsilon_k, \mathbf{z}_k)\right\rangle$$

$$= \|\mathbf{x}_{k-1} - \mathbf{x}\|^2 - \|\mathbf{x}_{k-1} - \mathbf{z}_k\|^2 - \|\mathbf{z}_k - \mathbf{x}_k\|^2 + 2\left\langle \mathbf{x} - \mathbf{z}_k, \eta\widehat{T}(\epsilon_k, \mathbf{z}_k)\right\rangle + 2\left\langle \mathbf{x}_k - \mathbf{z}_k, \mathbf{x}_{k-1} - \eta\widehat{T}(\epsilon_k, \mathbf{z}_k) - \mathbf{z}_k\right\rangle$$
$$\tag{8}$$

where (a) holds by using Fact 1. Note that

$$2\left\langle \mathbf{x}_* - \mathbf{z}_k, \eta\widehat{T}(\epsilon_k, \mathbf{z}_k)\right\rangle = 2\left\langle \mathbf{x}_* - \mathbf{z}_k, \eta\left(T(\mathbf{z}_k) + \epsilon_k\right)\right\rangle \leq 2\left\langle \mathbf{x}_* - \mathbf{z}_k, \eta\epsilon_k\right\rangle, \tag{9}$$

where the last inequality holds by the fact that $\langle \mathbf{x}_* - \mathbf{z}_k, T(\mathbf{z}_k)\rangle \leq 0$ since $\mathbf{x}_*$ is a solution of $\text{MVI}(T, \mathcal{X})$. Note that

$$2\left\langle \mathbf{x}_k - \mathbf{z}_k, \mathbf{x}_{k-1} - \eta\widehat{T}(\epsilon_k, \mathbf{z}_k) - \mathbf{z}_k\right\rangle$$

$$= 2\left\langle \mathbf{x}_k - \mathbf{z}_k, \mathbf{x}_{k-1} - \eta\widehat{T}(\epsilon_{k-1}, \mathbf{z}_{k-1}) - \mathbf{z}_k\right\rangle + 2\left\langle \mathbf{x}_k - \mathbf{z}_k, \eta\left(\widehat{T}(\epsilon_{k-1}, \mathbf{z}_{k-1}) - \widehat{T}(\epsilon_k, \mathbf{z}_k)\right)\right\rangle$$

$$\overset{(a)}{\leq} 2\eta\|\mathbf{x}_k - \mathbf{z}_k\| \cdot \left\|\widehat{T}(\epsilon_{k-1}, \mathbf{z}_{k-1}) - \widehat{T}(\epsilon_k, \mathbf{z}_k)\right\|$$

$$\overset{(b)}{\leq} 2\eta\left\|\Pi_{\mathcal{X}}\left(\mathbf{x}_{k-1} - \eta\cdot\widehat{T}(\epsilon_k, \mathbf{z}_k)\right) - \Pi_{\mathcal{X}}\left(\mathbf{x}_{k-1} - \eta\cdot\widehat{T}(\epsilon_{k-1}, \mathbf{z}_{k-1})\right)\right\| \cdot \left\|\widehat{T}(\epsilon_{k-1}, \mathbf{z}_{k-1}) - \widehat{T}(\epsilon_k, \mathbf{z}_k)\right\|$$

$$\overset{(c)}{\leq} 2\eta^2\left\|\widehat{T}(\epsilon_{k-1}, \mathbf{z}_{k-1}) - \widehat{T}(\epsilon_k, \mathbf{z}_k)\right\|^2 = 2\eta^2\|T(\mathbf{z}_{k-1}) + \epsilon_{k-1} - (T(\mathbf{z}_k) + \epsilon_k)\|^2$$

$$\leq 2\eta^2\left(\|T(\mathbf{z}_{k-1}) - T(\mathbf{z}_k)\| + \|\epsilon_{k-1}\| + \|\epsilon_k\|\right)^2 \overset{(d)}{\leq} 2\eta^2\left(L\|\mathbf{z}_{k-1} - \mathbf{z}_k\| + \|\epsilon_{k-1}\| + \|\epsilon_k\|\right)^2$$

$$\overset{(e)}{\leq} 6\eta^2\left(L^2\|\mathbf{z}_{k-1} - \mathbf{z}_k\|^2 + \|\epsilon_{k-1}\|^2 + \|\epsilon_k\|^2\right)$$
$$\tag{10}$$

where (a) holds by $\left\langle \mathbf{x}_k - \mathbf{z}_k, \mathbf{x}_{k-1} - \eta\widehat{T}(\epsilon_{k-1}, \mathbf{z}_{k-1}) - \mathbf{z}_k\right\rangle \leq 0$ and Cauchy-Schwartz inequality, where the former inequality comes from Fact 2 and the update rules of the algorithm, (b) holds by the update rule of $\mathbf{z}_k$ and $\mathbf{x}_k$, (c) holds by the nonexpansion property of the projection operator, (d) holds since $T$ is $L$-Lipschitz continuous, (e) holds since $(a + b + c)^2 \leq 3a^2 + 3b^2 + 3c^2$.

Define $\Lambda_k = 2\langle \mathbf{x}_* - \mathbf{z}_k, \eta\epsilon_k\rangle$. Taking $\mathbf{x} = \mathbf{x}_*$ in (8) and combining (9) and (10), we have

$$\|\mathbf{x}_k - \mathbf{x}_*\|^2$$
$$\leq \|\mathbf{x}_{k-1} - \mathbf{x}_*\|^2 - \|\mathbf{x}_{k-1} - \mathbf{z}_k\|^2 - \|\mathbf{z}_k - \mathbf{x}_k\|^2 + 6\eta^2 L^2\|\mathbf{z}_{k-1} - \mathbf{z}_k\|^2 + 6\eta^2\|\epsilon_{k-1}\|^2 + 6\eta^2\|\epsilon_k\|^2 + \Lambda_k$$
$$\tag{11}$$

Noting that

$$\|\mathbf{z}_{k-1} - \mathbf{z}_k\|^2 = \|\mathbf{z}_{k-1} - \mathbf{x}_{k-1} + \mathbf{x}_{k-1} - \mathbf{z}_k\|^2 \leq 3\|\mathbf{z}_{k-1} - \mathbf{x}_{k-1}\|^2 + 3\|\mathbf{x}_{k-1} - \mathbf{z}_k\|^2 + 3\|\mathbf{x}_k - \mathbf{z}_k\|^2,$$

we rearrange terms in (11), which yields

$$\|\mathbf{x}_{k-1} - \mathbf{z}_k\|^2 + \|\mathbf{z}_k - \mathbf{x}_k\|^2 - 6\eta^2 L^2\left(3\|\mathbf{z}_{k-1} - \mathbf{x}_{k-1}\|^2 + 3\|\mathbf{x}_{k-1} - \mathbf{z}_k\|^2 + 3\|\mathbf{z}_k - \mathbf{x}_k\|^2\right)$$
$$\leq \|\mathbf{x}_{k-1} - \mathbf{x}_*\|^2 - \|\mathbf{x}_k - \mathbf{x}_*\|^2 + 6\eta^2\|\epsilon_{k-1}\|^2 + 6\eta^2\|\epsilon_k\|^2 + \Lambda_k$$
$$\tag{12}$$

Take summation over $k = 1, \ldots, N$ in (12) and note that $\mathbf{x}_0 = \mathbf{z}_0$, which yields

$$
\left(1 - 18\eta^2 L^2\right) \sum_{k=1}^{N} \|\mathbf{x}_{k-1} - \mathbf{z}_k\|^2 + \left(1 - 36\eta^2 L^2\right) \sum_{k=1}^{N} \|\mathbf{x}_k - \mathbf{z}_k\|^2
$$

$$
\leq \|\mathbf{x}_0 - \mathbf{x}_*\|^2 - \|\mathbf{x}_N - \mathbf{x}_*\|^2 + 12\eta^2 \sum_{k=0}^{N} \|\epsilon_k\|^2 + \sum_{k=1}^{N} \Lambda_k
$$

(13)

By taking $\eta \leq \frac{1}{9L}$, we have $1 - 36\eta^2 L^2 \geq \frac{1}{2}$, and we have the result. $\qquad\square$

## C.3   MAIN PROOF OF THEOREM 1

*Proof.* Define $r_\eta(\mathbf{z}_k) = \|\mathbf{z}_k - \Pi_{\mathcal{X}}(\mathbf{z}_k - \eta T(\mathbf{z}_k))\|$. Our goal is to get a bound on $r_\eta(\mathbf{z}_k)$. We have:

$$
\begin{aligned}
r_\eta^2(\mathbf{z}_k) &= \|\mathbf{z}_k - \Pi_{\mathcal{X}}(\mathbf{z}_k - \eta T(\mathbf{z}_k))\|^2 = \|\mathbf{z}_k - \mathbf{x}_k + \mathbf{x}_k - \Pi_{\mathcal{X}}(\mathbf{z}_k - \eta T(\mathbf{z}_k))\|^2 \\
&\overset{(a)}{\leq} 2\|\mathbf{z}_k - \mathbf{x}_k\|^2 + 2\|\mathbf{x}_k - \Pi_{\mathcal{X}}(\mathbf{z}_k - \eta T(\mathbf{z}_k))\|^2 \\
&= 2\|\mathbf{z}_k - \mathbf{x}_k\|^2 + 2\left\|\Pi_{\mathcal{X}}\left(\mathbf{x}_{k-1} - \eta \widehat{T}(\epsilon_k, \mathbf{z}_k)\right) - \Pi_{\mathcal{X}}(\mathbf{z}_k - \eta T(\mathbf{z}_k))\right\|^2 \\
&\overset{(b)}{\leq} 2\|\mathbf{z}_k - \mathbf{x}_k\|^2 + 4\|\mathbf{x}_{k-1} - \mathbf{z}_k\|^2 + 4\eta^2 \left\|T(\mathbf{z}_k) - \widehat{T}(\epsilon_k, \mathbf{z}_k)\right\|^2 \\
&\leq 4\|\mathbf{z}_k - \mathbf{x}_k\|^2 + 4\|\mathbf{x}_{k-1} - \mathbf{z}_k\|^2 + 4\eta^2\|\epsilon_k\|^2
\end{aligned}
$$

(14)

where (a) holds since $(a + b)^2 \leq 2a^2 + 2b^2$, (b) holds by the non-expansion property of the projection operator and $(a + b)^2 \leq 2a^2 + 2b^2$.

Let $\mathbf{x}_* \in \mathcal{X}^*$, where $\mathcal{X}^*$ is the set of optimal solutions of MVI$(T, \mathcal{X})$, i.e. $\langle T(\mathbf{x}), \mathbf{x} - \mathbf{x}_* \rangle \geq 0$ holds for $\forall \mathbf{x} \in \mathcal{X}$. Define $\epsilon_k = \frac{1}{m_k} \sum_{i=1}^{m_k} T(\mathbf{z}_k, \xi_k^i) - T(\mathbf{z}_k)$, and $\widehat{T}(\epsilon_k, \mathbf{z}_k) = T(\mathbf{z}_k) + \epsilon_k$. Define $\Lambda_k = 2\langle \mathbf{x}_* - \mathbf{z}_k, \eta\epsilon_k \rangle$.

By summing over $k$ in Equation (14) and using Equation (7) in Lemma 1, we have

$$
\begin{aligned}
\sum_{k=1}^{N} r_\eta^2(\mathbf{z}_k) &\leq 4\sum_{k=1}^{N} \|\mathbf{z}_k - \mathbf{x}_k\|^2 + 4\sum_{k=1}^{N} \|\mathbf{x}_{k-1} - \mathbf{z}_k\|^2 + 4\eta^2 \sum_{k=1}^{N} \|\epsilon_k\|^2 \\
&= 8\left(\frac{1}{2}\sum_{k=1}^{N} \|\mathbf{z}_k - \mathbf{x}_k\|^2 + \frac{1}{2}\sum_{k=1}^{N} \|\mathbf{x}_{k-1} - \mathbf{z}_k\|^2\right) + 4\eta^2 \sum_{k=0}^{N} \|\epsilon_k\|^2 \\
&\overset{\text{By (7)}}{\leq} 8\left(\|\mathbf{x}_0 - \mathbf{x}_*\|^2 + 12\eta^2 \sum_{k=0}^{N} \|\epsilon_k\|^2 + \sum_{k=1}^{N} \Lambda_k\right) + 4\eta^2 \sum_{k=0}^{N} \|\epsilon_k\|^2
\end{aligned}
$$

(15)

Taking expectation and divided by $N$ on both sides, we have

$$
\begin{aligned}
\frac{1}{N}\sum_{k=1}^{N} \mathbb{E}\left[r_\eta^2(\mathbf{z}_k)\right] &\leq \frac{8}{N}\left(\|\mathbf{x}_0 - \mathbf{x}_*\|^2 + 12\eta^2 \sum_{k=0}^{N} \mathbb{E}\|\epsilon_k\|^2 + \sum_{k=1}^{N} \mathbb{E}(\Lambda_k)\right) + \frac{4\eta^2}{N}\sum_{k=1}^{N} \mathbb{E}\|\epsilon_k\|^2 \\
&\leq \frac{8}{N}\left(\|\mathbf{x}_0 - \mathbf{x}_*\|^2 + 12\eta^2 \sum_{k=0}^{N} \frac{\sigma^2}{m_k}\right) + \frac{4\eta^2}{N}\sum_{k=0}^{N} \frac{\sigma^2}{m_k} \\
&= \frac{8\|\mathbf{x}_0 - \mathbf{x}_*\|^2}{N} + \frac{100\eta^2}{N}\sum_{k=0}^{N} \frac{\sigma^2}{m_k}.
\end{aligned}
$$

(16)

$\qquad\square$

## D    PROOF OF THEOREM 2

In this section, we define $\mathbf{g}_k = T(\mathbf{z}_k)$, $\epsilon_k = \widehat{\mathbf{g}}_k - \mathbf{g}_k$.

### D.1    LEMMAS

**Lemma 2.** *For any positive definite diagonal matrix $H$ satisfying $H \succeq \delta I$ with $\delta > 0$, if $\|T(\mathbf{x}_1) - T(\mathbf{x}_2)\|_2 \leq L\|\mathbf{x}_1 - \mathbf{x}_2\|_2$ for $\mathbf{x}_1, \mathbf{x}_2 \in \mathcal{X}$, then*

$$\|T(\mathbf{x}_1) - T(\mathbf{x}_2)\|_{H^{-1}} \leq \frac{L}{\delta}\|\mathbf{x}_1 - \mathbf{x}_2\|_H.$$

*Proof.* Note that $H \succeq \delta I$, we have $0 < H^{-1} \preceq \frac{1}{\delta}I$. Noting that $\|\mathbf{x}\|_H = \sqrt{\mathbf{x}^\top H \mathbf{x}}$, we have

$$\|T(\mathbf{x}_1) - T(\mathbf{x}_2)\|_{H^{-1}} \leq \frac{1}{\sqrt{\delta}}\|T(\mathbf{x}_1) - T(\mathbf{x}_2)\|_2 \leq \frac{L}{\sqrt{\delta}}\|\mathbf{x}_1 - \mathbf{x}_2\|_2 \leq \frac{L}{\delta}\|\mathbf{x}_1 - \mathbf{x}_2\|_H.$$

$\square$

**Lemma 3.** *When $\eta \leq \frac{\delta}{9L}$, we have*

$$\frac{1}{2}\sum_{k=1}^{N}\|\mathbf{x}_{k-1} - \mathbf{z}_k\|_{H_{k-1}}^2 + \frac{1}{2}\sum_{k=1}^{N}\|\mathbf{x}_k - \mathbf{z}_k\|_{H_{k-1}}^2$$
$$\leq \sum_{k=1}^{N}\left(\|\mathbf{x}_{k-1} - \mathbf{x}_*\|_{H_{k-1}}^2 - \|\mathbf{x}_k - \mathbf{x}_*\|_{H_{k-1}}^2\right) + 12\eta^2\left(\|\epsilon_0\|_{H_0^{-1}}^2 + \sum_{k=1}^{N}\|\epsilon_k\|_{H_{k-1}^{-1}}^2\right) + \sum_{k=1}^{N}\Lambda_k \tag{17}$$

*Proof.* Define $\epsilon_k = \widehat{\mathbf{g}}_k - \mathbf{g}_k$. For any $\mathbf{x} \in \mathcal{X}$, we have

$$\|\mathbf{x}_k - \mathbf{x}\|_{H_{k-1}}^2 = \left\|\mathbf{x}_{k-1} - \eta H_{k-1}^{-1}\widehat{\mathbf{g}}_k - \mathbf{x}\right\|_{H_{k-1}}^2 = \left\|\mathbf{x}_{k-1} - \eta H_{k-1}^{-1}\widehat{\mathbf{g}}_k - \mathbf{x}\right\|_{H_{k-1}}^2 - \left\|\mathbf{x}_{k-1} - \eta H_{k-1}^{-1}\widehat{\mathbf{g}}_k - \mathbf{x}_k\right\|_{H_{k-1}}^2$$
$$= \|\mathbf{x}_{k-1} - \mathbf{x}\|_{H_{k-1}}^2 - \|\mathbf{x}_{k-1} - \mathbf{x}_k\|_{H_{k-1}}^2 + 2\left\langle \mathbf{x} - \mathbf{x}_k, \eta\widehat{\mathbf{g}}_k\right\rangle$$
$$= \|\mathbf{x}_{k-1} - \mathbf{x}\|_{H_{k-1}}^2 - \|\mathbf{x}_{k-1} - \mathbf{x}_k\|_{H_{k-1}}^2 + 2\left\langle \mathbf{x} - \mathbf{z}_k, \eta\widehat{\mathbf{g}}_k\right\rangle + 2\left\langle \mathbf{z}_k - \mathbf{x}_k, \eta\widehat{\mathbf{g}}_k\right\rangle$$
$$= \|\mathbf{x}_{k-1} - \mathbf{x}\|_{H_{k-1}}^2 - \|\mathbf{x}_{k-1} - \mathbf{z}_k + \mathbf{z}_k - \mathbf{x}_k\|_{H_{k-1}}^2 + 2\left\langle \mathbf{x} - \mathbf{z}_k, \eta\widehat{\mathbf{g}}_k\right\rangle + 2\left\langle \mathbf{z}_k - \mathbf{x}_k, \eta\widehat{\mathbf{g}}_k\right\rangle$$
$$= \|\mathbf{x}_{k-1} - \mathbf{x}\|_{H_{k-1}}^2 - \|\mathbf{x}_{k-1} - \mathbf{z}_k\|_{H_{k-1}}^2 - \|\mathbf{z}_k - \mathbf{x}_k\|_{H_{k-1}}^2 - 2\left\langle H_{k-1}(\mathbf{x}_{k-1} - \mathbf{z}_k), \mathbf{z}_k - \mathbf{x}_k\right\rangle +$$
$$\quad 2\left\langle \mathbf{x} - \mathbf{z}_k, \eta\widehat{\mathbf{g}}_k\right\rangle + 2\left\langle \mathbf{z}_k - \mathbf{x}_k, \eta\widehat{\mathbf{g}}_k\right\rangle$$
$$= \|\mathbf{x}_{k-1} - \mathbf{x}\|_{H_{k-1}}^2 - \|\mathbf{x}_{k-1} - \mathbf{z}_k\|_{H_{k-1}}^2 - \|\mathbf{z}_k - \mathbf{x}_k\|_{H_{k-1}}^2 + 2\left\langle \mathbf{x} - \mathbf{z}_k, \eta\widehat{\mathbf{g}}_k\right\rangle$$
$$\quad + 2\left\langle \mathbf{x}_k - \mathbf{z}_k, H_{k-1}(\mathbf{x}_{k-1} - \mathbf{z}_k) - \eta\widehat{\mathbf{g}}_k\right\rangle \tag{18}$$

Note that
$$2\left\langle \mathbf{x}_* - \mathbf{z}_k, \eta\widehat{\mathbf{g}}_k\right\rangle = 2\left\langle \mathbf{x}_* - \mathbf{z}_k, \eta\left(\mathbf{g}_k + \epsilon_k\right)\right\rangle \leq 2\left\langle \mathbf{x}_* - \mathbf{z}_k, \eta\epsilon_k\right\rangle, \tag{19}$$
where the last inequality holds by the fact that $\left\langle \mathbf{x}_* - \mathbf{z}_k, \mathbf{g}_k\right\rangle \leq 0$ since $\mathbf{x}_*$ is a solution of $\text{MVI}(T, \mathcal{X})$. Note that

$$2\left\langle \mathbf{x}_k - \mathbf{z}_k, H_{k-1}(\mathbf{x}_{k-1} - \mathbf{z}_k) - \eta\widehat{\mathbf{g}}_k\right\rangle$$
$$= 2\left\langle \mathbf{x}_k - \mathbf{z}_k, H_{k-1}(\mathbf{x}_{k-1} - \mathbf{z}_k - \eta H_{k-1}^{-1}\widehat{\mathbf{g}}_{k-1})\right\rangle + 2\left\langle \mathbf{x}_k - \mathbf{z}_k, \eta\left(\widehat{\mathbf{g}}_{k-1} - \widehat{\mathbf{g}}_k\right)\right\rangle$$
$$\overset{(a)}{\leq} 2\left\langle \left(\mathbf{x}_{k-1} - \eta H_{k-1}^{-1}\widehat{\mathbf{g}}_k\right) - \left(\mathbf{x}_{k-1} - \eta H_{k-1}^{-1}\widehat{\mathbf{g}}_{k-1}\right), \eta\left(\widehat{\mathbf{g}}_{k-1} - \widehat{\mathbf{g}}_k\right)\right\rangle$$
$$= 2\eta^2\|\widehat{\mathbf{g}}_{k-1} - \widehat{\mathbf{g}}_k\|_{H_{k-1}^{-1}}^2 = 2\eta^2\|\mathbf{g}_{k-1} - \mathbf{g}_k + \epsilon_{k-1} + \epsilon_k\|_{H_{k-1}^{-1}}^2$$
$$\overset{(b)}{\leq} 2\eta^2\left(\|\mathbf{g}_{k-1} - \mathbf{g}_k\|_{H_{k-1}^{-1}} + \|\epsilon_{k-1}\|_{H_{k-1}^{-1}} + \|\epsilon_k\|_{H_{k-1}^{-1}}\right)^2$$
$$\overset{(c)}{\leq} 2\eta^2\left(\frac{L}{\delta}\|\mathbf{z}_{k-1} - \mathbf{z}_k\|_{H_{k-1}} + \|\epsilon_{k-1}\|_{H_{k-1}^{-1}} + \|\epsilon_k\|_{H_{k-1}^{-1}}\right)^2$$
$$\overset{(d)}{\leq} 6\eta^2\left(\frac{L^2}{\delta^2}\|\mathbf{z}_{k-1} - \mathbf{z}_k\|_{H_{k-1}}^2 + \|\epsilon_{k-1}\|_{H_{k-1}^{-1}}^2 + \|\epsilon_k\|_{H_{k-1}^{-1}}^2\right) \tag{20}$$

where (a) holds by the update rule of $\mathbf{z}_k$ and $\mathbf{x}_k$ in Algorithm 2, (b) holds by the triangle inequality, (c) holds by utilizing the Lipschitz continuity of $T$, Lemma 2 and the fact that $H_{k-1} \succeq \delta I$ for any $k$, (d) holds since $(a+b+c)^2 \le 3a^2 + 3b^2 + 3c^2$.

Define $\Lambda_k = 2\langle \mathbf{x}_* - \mathbf{z}_k, \eta\epsilon_k \rangle$. Taking $\mathbf{x} = \mathbf{x}_*$ in (18) and combining (19) and (20), we have

$$\|\mathbf{x}_k - \mathbf{x}_*\|^2_{H_{k-1}} \le \|\mathbf{x}_{k-1} - \mathbf{x}_*\|^2_{H_{k-1}} - \|\mathbf{x}_{k-1} - \mathbf{z}_k\|^2_{H_{k-1}} - \|\mathbf{z}_k - \mathbf{x}_k\|^2_{H_{k-1}} + \frac{6\eta^2 L^2}{\delta^2}\|\mathbf{z}_{k-1} - \mathbf{z}_k\|^2_{H_{k-1}}$$
$$+ 6\eta^2\|\epsilon_{k-1}\|^2_{H_{k-1}^{-1}} + 6\eta^2\|\epsilon_k\|^2_{H_{k-1}^{-1}} + \Lambda_k \tag{21}$$

Noting that

$$\|\mathbf{z}_{k-1} - \mathbf{z}_k\|^2_{H_{k-1}} = \|\mathbf{z}_{k-1} - \mathbf{x}_{k-1} + \mathbf{x}_{k-1} - \mathbf{z}_k\|^2_{H_{k-1}}$$
$$\le 3\|\mathbf{z}_{k-1} - \mathbf{x}_{k-1}\|^2_{H_{k-1}} + 3\|\mathbf{x}_{k-1} - \mathbf{z}_k\|^2_{H_{k-1}} + 3\|\mathbf{z}_k - \mathbf{x}_k\|^2_{H_{k-1}},$$

we rearrange terms in (21), which yields

$$\|\mathbf{x}_{k-1} - \mathbf{z}_k\|^2_{H_{k-1}} + \|\mathbf{z}_k - \mathbf{x}_k\|^2_{H_{k-1}} - \frac{6\eta^2 L^2}{\delta^2}\left(3\|\mathbf{z}_{k-1} - \mathbf{x}_{k-1}\|^2_{H_{k-1}} + 3\|\mathbf{x}_{k-1} - \mathbf{z}_k\|^2_{H_{k-1}} + 3\|\mathbf{z}_k - \mathbf{x}_k\|^2_{H_{k-1}}\right)$$
$$\le \|\mathbf{x}_{k-1} - \mathbf{x}_*\|^2_{H_{k-1}} - \|\mathbf{x}_k - \mathbf{x}_*\|^2_{H_{k-1}} + 6\eta^2\|\epsilon_{k-1}\|^2_{H_{k-1}^{-1}} + 6\eta^2\|\epsilon_k\|^2_{H_{k-1}^{-1}} + \Lambda_k \tag{22}$$

Taking summation over $k = 1, \ldots, N$ in (22), and noting that $\mathbf{x}_0 = \mathbf{z}_0$, $\|\mathbf{x}\|^2_{H_{t-1}^{-1}} \ge \|\mathbf{x}\|^2_{H_t^{-1}}$ for all $\mathbf{x}$ and $t \ge 1$, we have

$$\left(1 - \frac{18\eta^2 L^2}{\delta^2}\right)\sum_{k=1}^{N}\|\mathbf{x}_{k-1} - \mathbf{z}_k\|^2_{H_{k-1}} + \left(1 - \frac{36\eta^2 L^2}{\delta^2}\right)\sum_{k=1}^{N}\|\mathbf{x}_k - \mathbf{z}_k\|^2_{H_{k-1}}$$
$$\le \sum_{k=1}^{N}\left(\|\mathbf{x}_{k-1} - \mathbf{x}_*\|^2_{H_{k-1}} - \|\mathbf{x}_k - \mathbf{x}_*\|^2_{H_{k-1}}\right) + 12\eta^2\left(\|\epsilon_0\|^2_{H_0^{-1}} + \sum_{k=1}^{N}\|\epsilon_k\|^2_{H_{k-1}^{-1}}\right) + \sum_{k=1}^{N}\Lambda_k \tag{23}$$

By taking $\eta \le \frac{\delta}{9L}$, we have $1 - \frac{36\eta^2 L^2}{\delta^2} \ge \frac{1}{2}$, and we have the result. $\qquad\square$

**Lemma 4.** *When* $\|\widehat{\mathbf{g}}_{1:N,i}\|_2 \le \delta N^\alpha$ *with* $0 \le \alpha \le 1/2$ *for every $i$, we have*

$$\sum_{k=1}^{N}\left(\|\mathbf{x}_{k-1} - \mathbf{x}_*\|^2_{H_{k-1}} - \|\mathbf{x}_k - \mathbf{x}_*\|^2_{H_{k-1}}\right) \le D^2\delta + D^2 \cdot d\delta(N-1)^\alpha \tag{24}$$

*Proof.*

$$\sum_{k=1}^{N}\left(\|\mathbf{x}_{k-1} - \mathbf{x}_*\|^2_{H_{k-1}} - \|\mathbf{x}_k - \mathbf{x}_*\|^2_{H_{k-1}}\right)$$
$$= \|\mathbf{x}_0 - \mathbf{x}_*\|^2_{H_0} - \|\mathbf{x}_1 - \mathbf{x}_*\|^2_{H_0} + \|\mathbf{x}_1 - \mathbf{x}_*\|^2_{H_1} - \|\mathbf{x}_2 - \mathbf{x}_*\|^2_{H_1} + \ldots + \|\mathbf{x}_{N-1} - \mathbf{x}_*\|^2_{H_{N-1}} - \|\mathbf{x}_N - \mathbf{x}_*\|^2_{H_{N-1}}$$
$$\le \|\mathbf{x}_0 - \mathbf{x}_*\|^2_{H_0} + \left(-\|\mathbf{x}_1 - \mathbf{x}_*\|^2_{H_0} + \|\mathbf{x}_1 - \mathbf{x}_*\|^2_{H_1}\right) + \ldots + \left(-\|\mathbf{x}_{N-1} - \mathbf{x}_*\|^2_{H_{N-2}} + \|\mathbf{x}_{N-1} - \mathbf{x}_*\|^2_{H_{N-1}}\right)$$
$$\le \|\mathbf{x}_0 - \mathbf{x}_*\|^2_{H_0} + D^2\left(\text{tr}(H_1 - H_0) + \text{tr}(H_2 - H_1) + \ldots + \text{tr}(H_{N-1} - H_{N-2})\right)$$
$$= \|\mathbf{x}_0 - \mathbf{x}_*\|^2_{H_0} + D^2\left(\text{tr}(H_{N-1} - H_0)\right) \le \|\mathbf{x}_0 - \mathbf{x}_*\|^2_{H_0} + D^2\text{tr}(H_{N-1}) \le D^2\delta + D^2 \cdot d\delta(N-1)^\alpha \tag{25}$$
$\qquad\square$

**Lemma 5.** *When* $\|\widehat{\mathbf{g}}_{1:N,i}\|_2 \le \delta N^\alpha$ *with* $0 \le \alpha \le 1/2$ *for every $i$, we have*

$$\mathbb{E}\left[96\eta^2\|\epsilon_0\|^2_{H_0^{-1}} + 100\eta^2\sum_{k=1}^{N}\|\epsilon_k\|^2_{H_{k-1}^{-1}}\right] \le \frac{96\eta^2\sigma^2}{m\delta} + 100\eta^2\left(2\delta dN^\alpha + \frac{G^2 d}{\delta}\right) \tag{26}$$

*Proof.* Note that

$$
\begin{aligned}
\mathbb{E}\left[\sum_{k=1}^{N}\|\epsilon_k\|_{H_{k-1}^{-1}}^2\right] &= \mathbb{E}\left[\sum_{k=1}^{N}\|\widehat{\mathbf{g}}_k - \mathbf{g}_k\|_{H_{k-1}^{-1}}^2\right] = \sum_{k=1}^{N}\mathbb{E}\|\widehat{\mathbf{g}}_k - \mathbf{g}_k\|_{H_{k-1}^{-1}}^2 = \sum_{k=1}^{N}\left(\mathbb{E}\|\widehat{\mathbf{g}}_k\|_{H_{k-1}^{-1}}^2 - \|\mathbf{g}_k\|_{H_{k-1}^{-1}}^2\right) \\
&\leq \sum_{k=1}^{N}\mathbb{E}\|\widehat{\mathbf{g}}_k\|_{H_{k-1}^{-1}}^2 = \sum_{k=1}^{N}\mathbb{E}\|\widehat{\mathbf{g}}_k\|_{H_k^{-1}}^2 + \sum_{k=1}^{N}\left(\mathbb{E}\|\widehat{\mathbf{g}}_k\|_{H_{k-1}^{-1}}^2 - \mathbb{E}\|\widehat{\mathbf{g}}_k\|_{H_k^{-1}}^2\right) \\
&= \sum_{k=1}^{N}\mathbb{E}\|\widehat{\mathbf{g}}_k\|_{H_k^{-1}}^2 + \sum_{k=1}^{N}\mathbb{E}\left\langle \widehat{\mathbf{g}}_k, (H_{k-1}^{-1} - H_k^{-1})\widehat{\mathbf{g}}_k\right\rangle \leq \sum_{k=1}^{N}\mathbb{E}\|\widehat{\mathbf{g}}_k\|_{H_k^{-1}}^2 + \sum_{k=1}^{N}\mathbb{E}\left[\mathrm{tr}(H_{k-1}^{-1} - H_k^{-1})G^2\right] \\
&\overset{(a)}{\leq} \mathbb{E}\left[2\sum_{i=1}^{d}\|\widehat{\mathbf{g}}_{1:N,i}\|_2\right] + \mathrm{tr}(H_0^{-1})G^2 \overset{(b)}{\leq} 2\delta d N^\alpha + \frac{G^2 d}{\delta}
\end{aligned}
\tag{27}
$$

where (a) holds since we have $\sum_{k=1}^{N}\|\widehat{\mathbf{g}}_k\|_{H_k^{-1}}^2 \leq 2\sum_{i=1}^{d}\|\widehat{\mathbf{g}}_{1:N,i}\|_2$ by the setting of $H_k$ and utilizing Lemma 4 of (Duchi et al., 2011), (b) holds because of $\|\widehat{\mathbf{g}}_{1:N,i}\|_2 \leq \delta N^\alpha$.

In addition, we have $\mathbb{E}\|\epsilon_0\|_{H_0^{-1}}^2 \leq \frac{\sigma^2}{m\delta}$, and hence

$$
\mathbb{E}\left[96\eta^2\|\epsilon_0\|_{H_0^{-1}}^2 + 100\eta^2\sum_{k=1}^{N}\|\epsilon_k\|_{H_{k-1}^{-1}}^2\right] \leq \frac{96\eta^2\sigma^2}{m\delta} + 100\eta^2\left(2\delta d N^\alpha + \frac{G^2 d}{\delta}\right)
\tag{28}
$$

$\square$

## D.2 MAIN PROOF OF THEOREM 2

*Proof.* Our goal is to bound $\frac{1}{N}\sum_{k=1}^{N}\mathbb{E}\|T(\mathbf{z}_k)\|_2^2$. Note that

$$
\begin{aligned}
\|\eta T(\mathbf{z}_k)\|_{H_{k-1}^{-1}}^2 &= \left\|H_{k-1}^{1/2}\left(\mathbf{z}_k - (\mathbf{z}_k - \eta H_{k-1}^{-1}T(\mathbf{z}_k))\right)\right\|^2 = \left\|H_{k-1}^{1/2}\left(\mathbf{z}_k - \mathbf{x}_k + \mathbf{x}_k - (\mathbf{z}_k - \eta H_{k-1}^{-1}T(\mathbf{z}_k))\right)\right\|^2 \\
&\overset{(a)}{\leq} \left(\left\|H_{k-1}^{1/2}(\mathbf{z}_k - \mathbf{x}_k)\right\| + \left\|H_{k-1}^{1/2}\left[\mathbf{x}_k - (\mathbf{z}_k - \eta H_{k-1}^{-1}T(\mathbf{z}_k))\right]\right\|\right)^2 \\
&\overset{(b)}{\leq} 2\left\|H_{k-1}^{1/2}(\mathbf{z}_k - \mathbf{x}_k)\right\|^2 + 2\left\|H_{k-1}^{1/2}\left[\mathbf{x}_k - (\mathbf{z}_k - \eta H_{k-1}^{-1}T(\mathbf{z}_k))\right]\right\|^2 \\
&\overset{(c)}{=} 2\|\mathbf{z}_k - \mathbf{x}_k\|_{H_{k-1}}^2 + 2\left\|H_{k-1}^{1/2}\left[\mathbf{x}_{k-1} - \eta H_{k-1}^{-1}\widehat{\mathbf{g}}_k - (\mathbf{z}_k - \eta H_{k-1}^{-1}T(\mathbf{z}_k))\right]\right\|^2 \\
&\overset{(d)}{\leq} 2\|\mathbf{z}_k - \mathbf{x}_k\|_{H_{k-1}}^2 + 4\|\mathbf{x}_{k-1} - \mathbf{z}_k\|_{H_{k-1}}^2 + 4\eta^2\|\widehat{\mathbf{g}}_k - T(\mathbf{z}_k)\|_{H_{k-1}^{-1}}^2 \\
&= 2\|\mathbf{z}_k - \mathbf{x}_k\|_{H_{k-1}}^2 + 4\|\mathbf{x}_{k-1} - \mathbf{z}_k\|_{H_{k-1}}^2 + 4\eta^2\|\epsilon_k\|_{H_{k-1}^{-1}}^2
\end{aligned}
\tag{29}
$$

where (a) holds by the triangle inequality, (b) is due to $(a+b)^2 \leq 2a^2 + 2b^2$, (c) holds by the update rule of $\mathbf{x}_k$ of Algorithm 2, (d) comes from the triangle inequality and $(a+b)^2 \leq 2a^2 + 2b^2$.

Taking summation over $k = 1, \ldots, N$ over (29) and invoking Lemma 3, we have

$$\sum_{k=1}^{N} \|\eta T(\mathbf{z}_k)\|_{H_{k-1}^{-1}}^2 \leq \sum_{k=1}^{N} \left( 2 \|\mathbf{z}_k - \mathbf{x}_k\|_{H_{k-1}}^2 + 4 \|\mathbf{x}_{k-1} - \mathbf{z}_k\|_{H_{k-1}}^2 + 4\eta^2 \|\epsilon_k\|_{H_{k-1}^{-1}}^2 \right)$$

$$\leq 8 \left( \frac{1}{2} \sum_{k=1}^{N} \|\mathbf{z}_k - \mathbf{x}_k\|_{H_{k-1}}^2 + \frac{1}{2} \sum_{k=1}^{N} \|\mathbf{x}_{k-1} - \mathbf{z}_k\|_{H_{k-1}}^2 \right) + 4\eta^2 \sum_{k=1}^{N} \|\epsilon_k\|_{H_{k-1}^{-1}}^2$$

$$\overset{\text{By (17)}}{\leq} 8 \left( \sum_{k=1}^{N} \left( \|\mathbf{x}_{k-1} - \mathbf{x}_*\|_{H_{k-1}}^2 - \|\mathbf{x}_k - \mathbf{x}_*\|_{H_{k-1}}^2 \right) + 12\eta^2 \left( \|\epsilon_0\|_{H_0^{-1}}^2 + \sum_{k=1}^{N} \|\epsilon_k\|_{H_{k-1}^{-1}}^2 \right) + \sum_{k=1}^{N} \Lambda_k \right)$$

$$+ 4\eta^2 \sum_{k=1}^{N} \|\epsilon_k\|_{H_{k-1}^{-1}}^2$$

$$= 8 \sum_{k=1}^{N} \left( \|\mathbf{x}_{k-1} - \mathbf{x}_*\|_{H_{k-1}}^2 - \|\mathbf{x}_k - \mathbf{x}_*\|_{H_{k-1}}^2 \right) + 96\eta^2 \|\epsilon_0\|_{H_0^{-1}}^2 + 100\eta^2 \sum_{k=1}^{N} \|\epsilon_k\|_{H_{k-1}^{-1}}^2 + 8 \sum_{k=1}^{N} \Lambda_k$$
$$(30)$$

Taking expectation on both sides, and invoking Lemma 4 and Lemma 5, and noting that $\mathbb{E}\left[ \sum_{k=1}^{N} \Lambda_k \right] = 0$, we have

$$\sum_{k=1}^{N} \mathbb{E} \|\eta T(\mathbf{z}_k)\|_{H_{k-1}^{-1}}^2 \leq 8 \left( D^2 \delta + D^2 \cdot d\delta(N-1)^\alpha \right) + \frac{96\eta^2 \sigma^2}{m\delta} + 100\eta^2 \left( 2\delta d N^\alpha + \frac{G^2 d}{\delta} \right) \quad (31)$$

Dividing $\eta^2 N$ on both sides, we have

$$\frac{1}{N} \sum_{k=1}^{N} \mathbb{E} \|T(\mathbf{z}_k)\|_{H_{k-1}^{-1}}^2 \leq \frac{8D^2 \delta^2 (1 + d(N-1)^\alpha)}{\eta^2 N} + \frac{100 \left( \sigma^2/m + d \left( 2\delta^2 N^\alpha + G^2 \right) \right)}{N} \quad (32)$$

$\square$

# E  MORE EXPERIMENTAL RESULTS ON CIFAR10

In Figure 7, we compare the performance of OSG, Alternating Adam (AlterAdam) and OAdagrad under the same minibatch size setting on CIFAR10 dataset, where one epoch means one pass of the dataset. We can see that OAdagrad and Alternating Adam behave consistently better than OSG. When the minibatch size is small (e.g., 64), OAdagrad and Alternating Adam have comparable performance, but when the minibatch size is large (e.g., 128, 256), OAdagrad converges faster than Alternating Adam. This phenomenon shows the benefits of OAdagrad when large minibatch size is used.

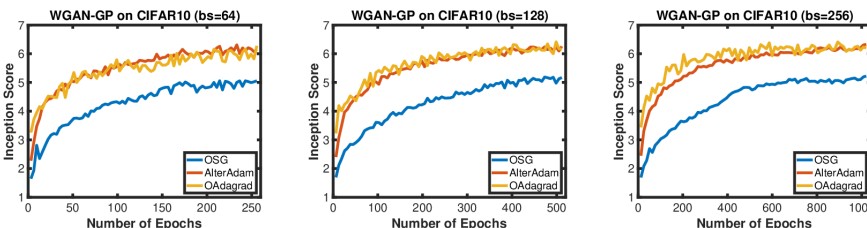

Figure 7: OAdagrad, OSG and Alternating Adam for WGAN-GP on CIFAR10 data with different batch sizes

## F THE EQUIVALENCE BETWEEN OSG IN UNCONSTRAINED CASE AND THE ALGORITHM IN DASKALAKIS ET AL. (2017)

Define $\hat{\mathbf{g}}_k = \frac{1}{m_k} \sum_{i=1}^{m_k} T(\mathbf{z}_k; \xi_k^i)$, then the update rule of Algorithm 1 becomes

$$\mathbf{z}_k = \mathbf{x}_{k-1} - \eta \hat{\mathbf{g}}_{k-1} \tag{33}$$

and

$$\mathbf{x}_k = \mathbf{x}_{k-1} - \eta \hat{\mathbf{g}}_k. \tag{34}$$

These two equalities together imply that

$$\mathbf{z}_{k+1} = \mathbf{x}_k - \eta \hat{\mathbf{g}}_k = \mathbf{x}_{k-1} - 2\eta \hat{\mathbf{g}}_k = \mathbf{z}_k + \eta \hat{\mathbf{g}}_{k-1} - 2\eta \hat{\mathbf{g}}_k, \tag{35}$$

where the first equality comes from (33) by replacing $k$ to $k+1$, the second equality holds by (34), and the third equality holds by using (33) again. (35) is the algorithm in (Daskalakis et al. 2017).

## G THE EXISTENCE OF MVI SOLUTION MAY NOT IMPLY PSEUDO-MONOTONICITY

Consider the function $f : \mathbb{R} \to \mathbb{R}$, where

$$f(x) = \begin{cases} \cos(x) & \text{if } 0 \leq x \leq 2\pi \\ 1 & \text{if } x \leq 0 \text{ or } x \geq 2\pi \end{cases}$$

Define $T(x) = \nabla f(x)$. Then $T(x) = -\sin(x)$ if $0 \leq x \leq 2\pi$ and $T(x) = 0$ if $x \leq 0$ or $x \geq 2\pi$. Then we know that $\pi$ is the solution of both SVI (i.e. $\langle T(\pi), x - \pi \rangle \geq 0$ for any $x \in \mathcal{X}$) and MVI (i.e. $\langle T(x), x - \pi \rangle \geq 0$ for any $x \in \mathcal{X}$). However $T$ is not pseudo-monotone. To see this, take $x = 0$ and $y = \frac{\pi}{4}$ and we have $\langle T(x), y - x \rangle = 0$ and $\langle T(y), y - x \rangle < 0$, which means that $T$ is not pseudo-monotone.

