# OpenReview forum: "Towards Better Understanding of Adaptive Gradient Algorithms in Generative Adversarial Nets"
_ICLR.cc/2020/Conference — Accept (Poster)_

### Official Review · AnonReviewer3 · 2019-10-20
**Official Blind Review #3**

**Rating:** 6

**Review:**

--------------------------
After the revisions made by the authors, my main concerns about the paper have been removed. Therefore, I am raising my score to 6 (weak accept).
--------------------------

Summary

The present work is concerned with the development of algorithms for the solution of variational inequalities in the stochastic setting, that is when the gradient computations are corrupted by noise. In this setting, the authors propose a variation of the extragradient method which they call Optimistic Stochastic Gradient (OSG), and show that by using a suitable method of variance reduction, the convergence rate of the algorithm matches the state of the art rate of convergence while relaxing the assumptions on the VI from pseudomonotone to assuming that the associated minty variational inequality has a solution. The authors furthermore introduce an adagrad-version of the same algorithm and show that improved convergence rates can be obtained depending on the growth rate of the cumulative stocastic gradients. An extensive suite of experiments studies the empirical performance of the proposed algorithms and compares them to the commonly used Adam optimizer.

Decision

In its present form, while some of the contributions of the paper seem to be relevant contributions (theoretical analysis of adagrad version, state of the art performance of optimistic mirror descent), other contributions (some of which are advertised strongly in the abstract) are incremental or implicit in earlier work (optimistic stochastic gradient descent, relaxation from pseudomonotone to minty VI). Furthermore, while the experimental part is detailed, the connection to the theory could arguably be strengthened more. I believe that by more concisely focusing the paper on its innovative aspects while relating it more directly to existing prior work, it could make for a much more valuable contribution to the literature, which is why I vote for rejecting the paper in its present form.

Additional Detail on decision

Novelty of OSG
The authors themselves note that the difference between OSG and OMD of Deskalakis et al is the inclusion of a projection step and the variance reduction by averaging multiple gradient evaluation at each iteration (which corresponds to choosing a different batch size for different iterations). I don't think these modifications are major enough to warrant "introduction of a new algorithm". The fact that all experiments seem to be conducted without constraints and constant batch size further strengthens this impression.

Relaxation to Minty VI
Another claimed improvement of the paper is relaxing the assumption of pseudomonotonicity to the mere assumption of existence of a variational inequality. However, if I'm not mistaken (please correct me if this assesment is incorrect), the only part where the pseudomonoonicity assumption enters the proof in Iusem et al is on page 36, to prove the last inequality of Equation (105). Here, however, the assumption of a Minty VI could equally be used. Thus, the weakened monotonicity assumption is not related to the use of SGO as opposed to extragradient, which is not apparent from the paper.

Suggestions for revision

I do think that the material can make for a solid paper, but I think it would strengthen rather than weaken the contribution to point out in more detail the close precursors of some of the results in the paper. Notably, rather than inventing a new algorithm (OSG), the paper proposes a way to combine variance reduction with Optimistic mirror descent of Deskalakis et al such as to achieve state of the art convergence rates which so far were only known for the variance reduced extragradient method. Furthermore, it points out that these convergence results (just as in the case of variance reduced extragradient) hold as soon as the associated Minty variational inequality holds true. This latter point would be much stronger, if examples were given that illustrate why this is a meaningful extension. Finally, the paper derives improved rates of an OAdagrad. In my opinion this point should be made more prominent by deemphasizing the part on OSG.

There is a typo in the definition of pseudomonotonicity.

Questions for authors

(1) Does figure 1 show iterations or epochs on the x axis? In order to support the theoretical claims, wouldn't it need to show the epochs on the x-axis? Otherwise, a larger batch-size simply corresponds to having access to more calls of the stochastic gradient oracle.

(2) Is there a new technical difficulty to overcome when replacing the extragradient method with the OSG compared to the proof of Iusem et al? If yes, can you give a concise description of it?

(3)It is not "evident from the definition" to me, why pseudo-monotonicity implies the existence of a solution of the Minty variational inequality, even though I believe that this is true. Could the authors explain this to me?

(4) The main improvement of OSG over extragradient lies in it needing to compute half the number of gradients per iteration. Is it possible to  explicitly compare the constants in the bounds on the convergence rates of the two algorithms? It seems that this would be required to truly make the case of a reduced complexity of OSG.



**Experience Assessment:**

I have published one or two papers in this area.

**Review Assessment: Checking Correctness Of Derivations And Theory:**

I assessed the sensibility of the derivations and theory.

**Review Assessment: Checking Correctness Of Experiments:**

I assessed the sensibility of the experiments.

**Review Assessment: Thoroughness In Paper Reading:**

I read the paper at least twice and used my best judgement in assessing the paper.

---

> ### Author Response · Authors · 2019-11-11
> **Thank you for your feedback. We updated the paper accordingly.**
>
> Thanks for your valuable comments. We have revised our paper according to your suggestions: 1) For OSG, we do not emphasize it is a new algorithm but rather emphasize our contribution on its analysis for general min-max problems. In the revision, we explicitly mention that OSG is not a new algorithm, which has been analyzed by Daskalakis et al. 2017 for the bilinear case. We emphasize that our work is the first to analyze OSG for a general min-max problem under the MVI assumption, and achieves the state-of-the-art results as stochastic extra-gradient method by Iusem et al. 2017. 2) We emphasize that the main difference between OSG and that of Iusem et al. 2017 is the number of stochastic gradient calculations per-iteration, and pointing out that the pseudomonotone assumption used by Iusem et al. 2017 can be replaced by our MVI assumption. 3) In page 4 remark point (a), we have clarified the relationship between the MVI assumption and the pseudomonotone assumption and also given an example that satisfies the MVI assumption but not the pseudomonotone assumption. 4) We emphasize that the main goal of this paper is to study the benefits of using adaptive gradient algorithms in training generative adversarial nets, and introducing its non-adaptive version (OSG) first is for the sake of comparison purpose, in order to highlight the benefits of adaptive methods in this context. The revisions have been highlighted in red.
>
>
> Q1: Does figure 1 show iterations or epochs on x-axis?
>
> A: The x-axis represents the number of iterations. We have included experiments with x-axis being the number of epochs at Figure 7 in the Appendix E (page 22) in the revised version.
>
>
> Q2: Is there a new technical difficulty to overcome when replacing the extragradient method with the OSG compared to the proof of Iusem et al? If yes, can you give a concise description of it?
>
> A: There are some subtle differences that requires deep investigation of OSG. One is the proof of Lemma 1 in Appendix C.2. Because of the replacement, the inequality (10) becomes different due to the different update, then we need to expand the term $\|z_{k-1}-z_k\|^2$ in a different way. The second one is in the proof of Theorem 1. Iusem et al. 2017 proved the convergence in terms of $x_k$, but we have to prove the convergence in terms of sequence $z_k$.
>
>
> Q3: Why pseudo-monotonicity implies the existence of a solution of the Minty variational inequality?
>
> A: This implication was not well explained in the manuscript, we corrected it. We assume that SVI has a nonempty solution set and that the operator T is pseudomonotone. Under those two assumptions there exists a solution for Minty VI (MVI).  We have made it more clear in the revised version (cf. Remark point (a) in page 4).
>
>
> Q4: Is it possible to explicitly compare the constants in the bounds on the convergence rate of the two algorithms?
>
> A: We did not make efforts to optimize the constants in the complexity of OSG. Currently, it is comparable to that but a little worse than (Iusem et al. 2017). In particular, our bound has a constant factor 4 (in our inequality (15)) while Iusem et al. 2017 has constant factor 2 (in their inequality (107)) when trying to bound the proximal gradient. However, our main goal is this paper is to demonstrate OAdaGrad could have a lower order of complexity in the presence of slow growth rate of cumulative stochastic gradient.

---

> > ### Comment · AnonReviewer3 · 2019-11-14
> > **Thank you for the revisions**
> >
> > Thank you making the effort to answer my questions and suggestions for revision.
> >
> > A minor comment at this point: On page 8 of the new version you write
> > "Figure 1 suggests
> > that OAdagrad performs better than OSG and Alternating Adam, and OAdagrad results in higher IS.
> > In addition, for OSG and OAdagrad, large batch size helps the training, which is consistent with our
> > theory."
> >
> > I am not convinced how closely the fact that you described is related to you theory. Even without any theory it seems very natural that more accurate gradient estimates will improve convergence speed and I don't see how the more subtle results of your theory are confirmed by those experiments.
> > Please let me know if you disagree but otherwise I would maybe advise to drop this remark.

---

> > > ### Author Response · Authors · 2019-11-14
> > > **Thank you for your suggestions.**
> > >
> > > Thank you for pointing it out. We agree with it and have dropped this remark.

---

### Official Review · AnonReviewer2 · 2019-10-23
**Official Blind Review #2**

**Rating:** 6

**Review:**

This paper proposed a new algorithm (Optimistic Stochastic Gradient) for solving a class of non-convex non-concave min-max problem. The convergence theory is established for finding first order stationary point. The authors also proposed an adaptive variant of the proposed algorithm, called OAdagrad and showed an improved adaptive complexity.

- It is not immediately clear to me why the update of Algorithm 1 becomes the algorithm in (Daskalaskis et al. 2017) when there is no constraint. Can the authors further explain this clearly? The two variables in (Daskalaskis et al. 2017) are updated with different signs, but here  z=(u,v)  (I assume, the authors should clearly define z too), it seems that u and v are updated with the same sign?

- In terms of the theoretical contributions, aside from the presenting theorems, can the authors also comment on what is the key point to achieve the derived results? For example, in Algorithm 1, the only difference from stochastic extragradient method is T(z_{k-1}) instead of T(x_{k-1}), and the theorem achieves the same iteration complexity with weaker assumption.  Is this because the replacement term, or it is the proving technique improvement that can also be applied to stochastic extragradient method?

- For Algorithm 2, why there is no projection operator like in Algorithm 1? Also, the algorithm design is a bit different from traditional AdaGrad as their H matrix is the historical average of all past gradient square (element-wise). Here the s_i is L_2 norm of historical gradient, there is no averaging (divided by k) operator. Can the authors elaborate on this algorithm design?

- For experiments part, it is better for the authors to compare with more baseline methods such as Optimistic Adam. For Figure 1, can the authors put different batches in different plot so that we can directly compare the performances of different algorithms? It does not make sense that alternating Adam failed in training SA-GAN here but success in the original paper. Can the authors figure it out and provide the comparison on this?

Detailed comments:

- it is better to define m_t in or before Algorithm 1 to make it clear for the readers.
- In Page 7, comparison with … paragraph, should be \beta_1 = 0, \beta_2 -> 1? If so, it is still a bit different with OAdagrad?


======================
after the rebuttal

I thank the authors for their response and it addressed most of my concerns.

**Experience Assessment:**

I have published one or two papers in this area.

**Review Assessment: Checking Correctness Of Derivations And Theory:**

I assessed the sensibility of the derivations and theory.

**Review Assessment: Checking Correctness Of Experiments:**

I carefully checked the experiments.

**Review Assessment: Thoroughness In Paper Reading:**

I read the paper at least twice and used my best judgement in assessing the paper.

---

> ### Author Response · Authors · 2019-11-11
> **2/2 Thank you for your feedback, we included updates and clarifications.**
>
> Q4: For Algorithm 2, the algorithm design is a bit different from traditional Adagrad as their H matrix is the historical average of all past gradient square (element-wise). Here the s_i is L_2 norm of historical gradient, there is no averaging (divided by k) operator. Can the authors elaborate on this algorithm design?
>
> A: It is indeed the same as traditional Adagrad. Please note that in the original Adagrad paper (Duchi et al, 2011), there are two variants (in their Figure 1): the first one is using primal-dual subgradient update, and the second one is using composite mirror descent update. The first one utilizes the averaging operator while the second one does not. Our algorithm design is inspired by the second variant.
>
> Q5: Experiments.
>
> A:
> (1) The reason why we didn't compare with Optimistic Adam was that our algorithm is a special case of Optimistic Adam with $\beta_1=0$ and $\beta_2$ approaches to $1$. We mentioned this in the paragraph above section 5 at Page 7. The authors of the original Adam paper also make the similar claim.
>
> (2) We have added another Figure 7 in Appendix E (page 22) in the revision to put different batches in different plots to compare algorithms. Thanks for the suggestion.
>
> (3) The reason why alternating Adam failed in our SA-GAN experiment was that we used different batch size (128) compared with the original paper (256). Due to limited computational resources, we only used 2 machines with minibatch 64 for each machine, while they used 4 machines in SA-GAN paper (with 64 minibatch per machine). It appears that using the specific range of learning rates we tried didn't work out very well. We noticed that Alternating Adam was very sensitive to the choice of learning rates and we did not manage to get Alternating Adam to work for our particular batch size and for the ranges of learning rates we tried. This does not mean that alternating Adam fails, it just needs more tuning to find the correct range of learning rates for the particular batch size we have. This is in line with the conclusions of the paper “are all GANs created equal? A large scale study ” https://arxiv.org/abs/1711.10337 , that shows that the computational budget dedicated to tuning the hyperparameters in the GAN context is crucial to their performance. We clarified this in the paper (In experiment Section page 9).
>
> Note that we presented results of Alternating Adam on CIFAR 10 where the training succeed.
>
>
> Q6: Detailed comments.
>
> A: Thank you for carefully reading our paper! We have defined $m_t$ and make it more clear in the revision. In terms of the paragraph in page 7, you are absolutely right, it should be $\beta_1=0, \beta_2\rightarrow1$. As we mentioned in Q5(1) and also in the paragraph, in this case it is equivalent to OAdagrad.

---

> ### Author Response · Authors · 2019-11-11
> **1/2 Thank you for your feedback, we included updates and clarifications.**
>
> Thanks for your valuable comments. We have made revisions according to your suggestions in red text.
>
> Q1: Why the update of Algorithm 1 becomes the algorithm in (Daskalaskis et al. 2017) when there is no constraint? It seems that u and v are updated with the same sign?
>
> A: $u$ and $v$ are updated with different signs due to the definition of operator $T$, which is defined as $T(x)=[\nabla_u F(u,v), -\nabla_v F(u,v)]^\top$ (please note that there is a negative sign before $\nabla_v F(u,v)$). Let's clarify the equivalence of the update rule between Algorithm 1 and the algorithm in (Daskalakis et al. 2017) when there is no constraint. Define $\hat{g}_k = \frac{1}{m_k}\sum_{i=1}^{m_k}T(z_k;\xi_k^i)$, then the update rule of Algorithm 1 becomes $z_k=x_{k-1}-\eta \hat{g}_{k-1}$ (i) and $x_k=x_{k-1}-\eta \hat{g}_k$ (ii). These two together imply that $z_{k+1}=x_k-\eta \hat{g}_k=x_{k-1}-2\eta \hat{g}_k=z_k+\eta \hat{g}_{k-1}-2\eta \hat{g}_k$ (iii), where the first equality comes from (i) by replacing $k$ to $k+1$, the second equality holds by (ii), and the third equality holds by using (i) again. (iii) is the algorithm in (Daskalakis et al. 2017).
>
> We have added the details in the supplement in the revision for your information (Section F in the appendix) .
>
>
> Q2: In terms of theoretical contributions, what is the key point to achieve the derived results? For example, why the theorem for Algorithm 1 achieves the same iteration complexity with weaker assumption?
>
> A: For Algorithm 1, the main difference from extra-gradient (Iusem et al 2017) is that Algorithm 1 only requires computing stochastic gradient once instead of twice in each iteration. For theoretical analysis, the key point is that we use improved analysis over that of Iusem et al 2017, which can tolerate the error term induced by the replacement without hurting the iteration complexity. Indeed, we have realized that Iusem et al’ analysis can hold under the same MVI assumption. For Algorithm 2, the novelty is to use an adaptive step size and its proof is much more involved in order to dealing with the complex adaptive step size. We not only show the improved adaptive complexity of Algorithm 2 but also eliminate the large minibatch size requirement as needed in Algorithm 1. The key point to derive Theorem 2 is to carefully design a variable-metric which adapts to the geometry of data during the update to get potentially faster convergence (Lemma 5) while controlling the error term not to blow up (Lemma 3, Lemma 4, Lemma 6) simultaneously.
>
>
> Q3: For Algorithm 2, why there is no projection operator like in Algorithm 1?
>
> A: In this paper, we only analyze Algorithm 2 for the unconstrained case, i.e. $X = R^d$. We have made it more clear in the revised version. Thanks for pointing it out.

---

### Official Review · AnonReviewer1 · 2019-10-23
**Official Blind Review #1**

**Rating:** 6

**Review:**

This paper proposes two methods named OSG and OAdagrad for solving stochastic non-convex non-concave min-max problems. In theoretical analyses, a convergence rate of $O(\epsilon^{-4})$ and a much better rate are provided for OSG and OAdagrad, respectively, to find the $\epsilon$-accurate first-order stationary point. Finally, the superior performance of the proposed method is empirically verified on training generative adversarial networks (GANs).

Clarity:
The paper is well organized and easy to read.

Quality:
The work is of good quality and is technically sound. However, I did not verify the proof in detail.

Significance:
A stochastic non-convex non-concave minimax problem studied in this paper is recently considered as an important class of the optimization problems because important machine learning problems such as GANs fall into this class and most past papers studied convex-concave min-max problems instead. A few studies [Iusem+(2017), Lin+(2018)] proposed optimization algorithms for this problem and derived convergence rates $O(\epsilon^{-4})$ and $O(\epsilon^{-6})$, respectively. On the other hand, proposed methods have several preferable properties compared to these methods. For instance, OSG exhibits a comparable convergence rate to [Iusem+(2017)] with a fewer per-iteration complexity and OAdagrad exhibits a much faster convergence rate $O(\epsilon^{-2/(1-\alpha)})$ depending on the parameter $\alpha$ that is an order of the growth of the cumulative stochastic gradients norm. In addition, the order of $\alpha$ is shown to be slow in general and certainly faster convergence rates are also confirmed in experiments on training GANs. Thus, experimental results seem consistent with the theory.
Since a derived convergence rate of OAdagrad is potentially much faster than those of existing methods, I think the OAdagrad is one of the promising methods for training GANs.

**Experience Assessment:**

I have read many papers in this area.

**Review Assessment: Checking Correctness Of Derivations And Theory:**

I did not assess the derivations or theory.

**Review Assessment: Checking Correctness Of Experiments:**

I assessed the sensibility of the experiments.

**Review Assessment: Thoroughness In Paper Reading:**

I made a quick assessment of this paper.

---

> ### Author Response · Authors · 2019-11-11
> **Thank you for your review.**
>
> Thanks for your encouraging comments and your constructive feedback.

---

### Author Response · Authors · 2019-11-11
**General Comments**

Thanks for all the comments. We have updated our manuscript per reviewers’ suggestions. All updates are marked in red. The main summary of the updates are:

1. We change the statement in terms of OSG in both abstract and the claim of main contribution, as suggested by R3. We put our work in the context of previous works when introducing OSG and draw connections to prior works (e.g. Daskalakis et al. 2017, Iusem et al. 2017).

2. Per R3’s suggestion, we mention that our MVI assumption is also sufficient to derive results in (Iusem et al. 2017) in Table 1. The MVI assumption and pseudo-monotonicity are clarified, and an example that satisfies MVI assumption but not pseudo-monotonicity is also provided (cf. Remark point (a) in page 4).

3. We explain the equivalence between OSG and the algorithm in (Daskalakis et al. 2017) in Appendix F, as suggested by R2.

4. We have added another Figure 7 in Appendix E (page 22) in the revision to put different batches in different plots to compare algorithms. The figures show the inception score versus the number of epochs, as suggested by R2 and R3.

---

### Decision · Program_Chairs · 2019-12-19

**Decision:**

Accept (Poster)

**Comment:**

This work proposes a new adaptive method for solving certain min-max problems.

The reviewers all appreciated the work and most of their concerns were addressed in the rebuttal. Given the current interest in both adaptive methods and min-max problems, this work is suited for publication at ICLR.